# SEARCHING FOR HIGH-VALUE MOLECULES USING REINFORCEMENT LEARNING AND TRANSFORMERS

**Raj Ghugare**[1,2]     **Santiago Miret**[3]     **Adriana Hugessen**[1,2]
**Mariano Phielipp**[3]     **Glen Berseth**[1,2]

[1]Université de Montréal     [2]Mila - Quebec AI Institute     [3]Intel Labs

raj.ghugare@mila.quebec , santiago.miret@intel.com,
adriana.knatchbull-hugessen@mila.quebec

## ABSTRACT

Reinforcement learning (RL) over text representations can be effective for finding high-value policies that can search over graphs. However, RL requires careful structuring of the search space and algorithm design to be effective in this challenge. Through extensive experiments, we explore how different design choices for text grammar and algorithmic choices for training can affect an RL policy's ability to generate molecules with desired properties. We arrive at a new RL-based molecular design algorithm (ChemRLformer) and perform a thorough analysis using 25 molecule design tasks, including computationally complex protein docking simulations. From this analysis, we discover unique insights in this problem space and show that ChemRLformer achieves state-of-the-art performance while being more straightforward than prior work by demystifying which design choices are actually helpful for text-based molecule design.

## 1 INTRODUCTION

Molecular discovery can have a significant impact on our society, however, the vast search space makes it challenging to find high-value molecules. The potential of reinforcement learning (RL) methods to discover new, high-value molecules has resulted in a series of research work performed by RL researchers focusing on learning policies as graph neural networks (GNNs) (You et al., 2018; Zhou et al., 2019; Jin et al., 2020; Fu et al., 2022; Yang et al., 2021; Bengio et al., 2021). In this formulation, the RL policy is trained to add atoms and bonds to a molecular graph representation. In this formulation there is a one-to-one mapping between molecules and their graph representation, making it easier to construct state and action spaces with Markovian dynamics. However, the action space in the graph formulation is vast as it consists of the product of candidate attachment positions and candidate attachment sequences. Graph-based data structures (such as adjacency matrices, trees, etc.) are a powerful representation used to describe a number of design problems, including social networks (Tan et al., 2019), transportation networks (Wang and Tang, 2021), recommendation systems (Chen et al., 2021b), and combinatorial optimization problems (Khadka et al., 2020; Miret et al., 2022) have been popular in this design space. However, GNNs are often difficult to train (Chen et al., 2022) and cannot readily take advantage of large-scale text data sets that effectively describe molecular structures and properties.

In order to take advantage of the richness of text-based representations for molecules, one can formulate the molecular search problem as the construction of tokens in a sequence that become a molecular text. The molecular texts formulated by common text-based representations, such as SMILES (Weininger, 1988) and SELFIES (Krenn et al., 2020), can then be converted into molecular graphs with cheminformatics libraries (Landrum et al., 2013) using their respective encoding and decoding rules. However, the text-based representation can be more difficult to formulate as an MDP since there is not always an exact one-to-one mapping between texts and molecules. In fact, the text-to-molecule conversion can be many-to-one, where the complexity of the dynamics in the MDP given by many-to-one mappings is non-trivial. On the other hand, the action space in molecular text design can be significantly reduced given the rules of text construction imposed by a given representation. Moreover, formulating molecule discovery as sequence-generation has the potential to capitalize on recent successes in natural language modeling (Brown et al., 2020a).

In this paper, we perform a detailed empirical study of molecular discovery using text-based RL across more than 25 molecular properties relevant for drug-discovery, including docking simulation for molecular ligands (García-Ortegón et al., 2022; Huang et al., 2022) and develop our own algorithm (ChemRLformer) based on state-of-the-art literature as shown in Table 1. In our experiments, we evaluate two molecular text representations (SMILES, SELFIES) and the use of three neural network architectures (Multi-Layer Perceptron (Bengio et al., 2003), Recurrent Neural Network (Schmidt, 2019), Transformer (Vaswani et al., 2017)) pretrained on 5 datasets of varying quality and sizes. We create ChemRLformer that achieves the highest performance across these tasks while being much simpler than previous text-based RL algorithms (Blaschke et al., 2020a; Gao et al., 2022b). Via our detailed ablation study, we construct ChemRLformer and find that pretraining on *aligned* datasets can significantly improve performance across all molecular design tasks, even exceeding the performance of agents pretrained on 100 times larger datasets. We also show that targeted algorithmic design, such as hill-climbing in the replay buffer and regularization, further increases the performance of ChemRLformer. To the best of our knowledge, ChemRLformer is the largest analysis of text-based RL methods for molecule discovery.

Table 1: Table showing conceptual comparisons of various text based molecular optimization methods. ChemRLformer combines the most successful elements of prior work.

| Method | Text Representation | RL | Architecture | Pretraining | Algorithmic Components |
|---|---|---|---|---|---|
| SMILES-VAE (Gómez-Bombarelli et al., 2018) | SMILES | ✗ | VAE | ✓ | Maximum Likelihood |
| SMILES-LSTM (Brown et al., 2019) | SMILES | ✗ | LSTM | ✓ | Maximum Likelihood |
| BOSS (Moss et al., 2020) | SMILES | ✗ | VAE | ✗ | Bayesian Optimization |
| REINVENT (Blaschke et al., 2020a) | SMILES | ✓ | GRU | ✓ | Replay buffer, KL |
| REINVENT 2.0 (Blaschke et al., 2020b) | SMILES | ✓ | GRU | ✓ | HC-Replay buffer, Log p, KL |
| Taiga 2.0 (Mazuz et al., 2023) | SMILES | ✓ | TRANSFORMER | ✓ | policy gradients |
| MolGPT (Bagal et al., 2021) | SMILES | ✓ | TRANSFORMER | ✓ | policy gradients |
| STONED (Nigam et al., 2021) | SELFIES | ✗ | FC | ✗ | Genetic algorithm |
| Pasithea (Shen et al., 2021) | SELFIES | ✗ | FC | ✗ | Deep dreaming |
| **ChemRLformer (Ours)** | SMILES, SELFIES | ✓ | Transformer, FC | ✓ | Replay buffer, KL |

## 2 RELATED WORK

**RL for Design and Discovery**: Many methods in diverse fields leverage RL to help augment a prior design method to improve performance (Yu et al., 2018; Schaff et al., 2019). Other methods have explicitly included the design process in the RL loop by training design problems together (Chen et al., 2021a; Ha, 2019; Luck et al., 2020; Kumar et al., 2022) with most prior work focusing on robot and agent design, not molecular design. Our molecular design work creates an autoregressive structure that grows the size of the state as the agent acts in the environment.

**Molecular Discovery Using Sequence-Based Methods:** Sequence-based methods treat molecular design as a sequence of tokens that get concatenated in order. Generative models for sequence-based methods span a diverse range, including variational autoencoders (VAEs) (Gómez-Bombarelli et al., 2018; Alperstein et al., 2019), recurrent neural networks (RNNs) (Gupta et al., 2018; Bjerrum and Threlfall, 2017; Grisoni et al., 2020; Flam-Shepherd et al., 2022) and transformer models(Wang et al., 2019; Fabian et al., 2020; Edwards et al., 2022a; Zeng et al., 2022; Taylor et al., 2022). The general procedure for all the above methods is to perform self-supervised generative learning to sample molecules similar to the original dataset. MoLRL can also make use of pretrained generative models, which we then fine-tune using reinforcement learning to produce enhanced molecules.

**Molecular Discovery Using Search-Based Methods:** Although sequence-based molecule generation methods often provide a more structured way of learning molecular distributions, search-based methods generally have the advantage of being able to directly find molecules based on a desired property. Although a wide range of graph-based RL methods (You et al., 2018; Zhou et al., 2019; Jin et al., 2020; Fu et al., 2022; Yang et al., 2021; Bengio et al., 2021) for optimizing molecules exist, graph-based state representations introduce significant complexity to the RL problem formulation, both in the transition dynamics and action space. By contrast, text-based methods are simpler and also relatively under-explored, motivating our focus on these methods in this work. Moreover, recent work (Cieplinski et al., 2021; Gao et al., 2022b) has shown that an older text-based method REINVENT (Olivecrona et al., 2017) outperforms more complex graph-based RL methods. Some limited extensions to Olivecrona et al. (2017) have been explored, including

experimenting with a newer molecular grammar designed for robust molecule generation (Gao et al., 2022b). However, there has been limited work proposing the use of language models and text-based RL for molecular discovery. Additionally, there have been limited efforts to incorporate recent advancements from the language modeling domain into these methods. For example, the a character-level LSTM network architecture used in Olivecrona et al. (2017), has not been revisited despite significant recent advances in sequence modeling (Vaswani et al., 2017; Brown et al., 2020b).

## 3 BACKGROUND

The algorithms detailed in this paper are built on top of a foundation of reinforcement learning, text-based molecule representations, and language modeling.

**Reinforcement Learning:** Reinforcement learning can be used to learn policies for sequential decision-making problems. Policies are optimized based on an environment that is described as a Markov Decision Process (MDP). A discrete MDP is defined by the tuple $\langle \mathcal{S}, \mathcal{A}, \mathcal{T}, r, \gamma \rangle$ where $\mathcal{S}$ is the state space, $\mathcal{A}$ is the action space, $T : \mathcal{S} \times \mathcal{A} \times \mathcal{S}' \to [0, 1]$ is the transition function, $r : \mathcal{S} \times \mathcal{A} \to \mathcal{R}$ is the reward function and $\gamma$ is the discount rate.

For actions $a_t \in \mathcal{A}$ and states $s_t \in \mathcal{S}$, the goal of reinforcement learning is to learn a policy $\pi_\theta(a_t|s_t)$ which maps states to actions, such that:

$$\pi_\theta(a_t|s_t) = \arg \max_\theta \mathbf{E}_{p(\tau|\theta)} \left[ \sum_{t=0}^{T} \gamma^t r(s_t, a_t) \right] \tag{1}$$

where $p(\tau|\theta)$ is the distribution over trajectories induced by $\pi_\theta$ and the transition function $\mathcal{T}$.

**Text representations for molecules:** Molecules are most naturally described using a graph structure of atoms and bonds. However, graph-based deep learning models can be difficult to train, especially at large scale (Dwivedi et al., 2022; Geisler et al., 2023). Recent works have proposed a variety of text representations for molecules (Weininger, 1988; Krenn et al., 2020; Heller et al., 2013; Krenn et al., 2022; Cheng et al., 2023), each having their distinct advantages and shortcomings. In this study, we focus on the two most commonly used representations: SMILES (Weininger, 1988) and SELFIES (Krenn et al., 2020). Any text representation for molecules consists of a set of valid tokens, which may represent individual atoms or special characters that imply the presence of certain structures, as well as the encoding and decoding rules needed to convert between the text representation and the graph representation of a molecule. Valid texts under a grammar are those which respect both the vocabulary and the encoding/decoding rules for that grammar and, hence, can be converted into a graph representation of a molecule. SELFIES, which was developed in response to the tendency for SMILES-based deep learning models (Gó mez-Bombarelli et al., 2018; Jin et al., 2018) to generate invalid molecular texts, has the useful property of providing a conversion for *any* text into a graph corresponding to a molecule, provided the tokens in the text respect the SELFIES vocabulary. For example, the text representation of Benzene in SMILES is C1=CC=CC=C1 while in SELFIES one possible representation is [C][=C][C][=C][C][=C][Ring1][=Branch1].

**Language modeling:** Language modeling often relies on the self-supervised task of next-token prediction for model pretraining. The general framework for next-token prediction is to train a model to predict the next token in a sequence autoregressively, i.e. given the previous tokens in the sequence (left context). Many architectures to handle sequential data have been proposed: Recurrent Neural Networks (RNNs) (Hochreiter and Schmidhuber, 1997; Rumelhart and McClelland, 1987) are a class of models used in sequence modeling which use recursive connections in hidden layers to accumulate the left context for next-token prediction. Transformers are a more recent architecture that instead use a self-attention mechanism (Vaswani et al., 2017) to capture dependencies between all tokens in a sequence. For next-token prediction tasks, attention masking is used to enforce left context, meaning that representations for tokens later in the sequence are only allowed to attend to previous tokens in the sequence. In Section 4 we outline how we pretrain an autoregressive sequence model to predict sequences of known molecules.

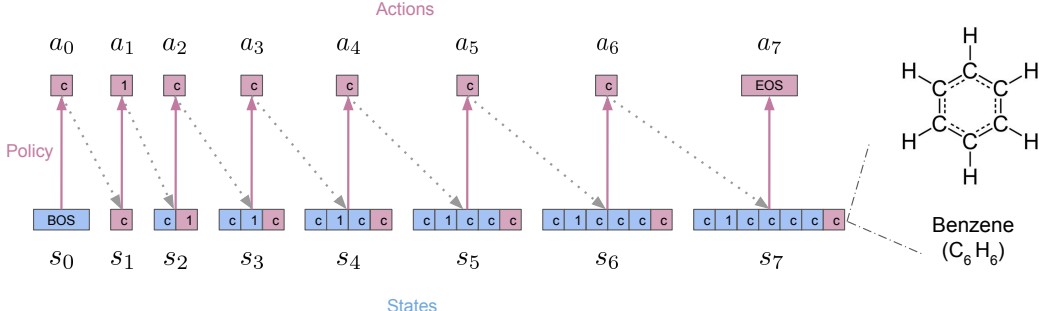

Figure 1: **Autoregressively generating a benzene molecule.**: An autoregressive model for sequence generation can be viewed as an RL policy where the actions $a_t$ are the next tokens to append to the sequence and the state is the concatenation of all actions taken up to time $t - 1$. A special end-of-sequence token can terminate the episode early at time $T$. The text at time $s_T$ is then converted into a molecule based on the text-representation grammar and then scored according to a scoring function that measures the alignment of the molecule with the desired properties informed by the application. Hydrogen atoms are added at the end to complete the structure.

## 4 CHEMRLFORMER GENERATING MOLECULAR STRINGS VIA REINFORCEMENT LEARNING

The space of drug-like molecules is vast, and reinforcement learning methods hold great promise in improving the speed and reducing the cost of drug discovery. In this section, we describe ChemRLformer and how combining language models and tools from RL produces a sota algorithm.

**MDP for molecule generation:** The vocabulary and grammar for text representations of molecules can be interpreted as an MDP as described in Section 3, where the states $s_t$ correspond to a *variable length* text of accumulating tokens, and the actions $a_t$ correspond to vocabulary defined by the text-representation. The transition function is a deterministic function where the action $a_t$ taken by the agent is appended to the end of the state $s_t$ resulting in $s_{t+1}$ using the dynamics $s_{t+1} = [s_t, a_t] \leftarrow T(s_t, a_t)$. However, the corresponding transition function induced in the graph representation of molecules is more complex as it is determined by the encoding/decoding rules of the chosen text representation. For example, in the SMILES grammar, a random concatenation of tokens may not correspond to a valid molecule, while the SELFIES grammar is constructed such that any ordering of its tokens is encoded as a valid molecule.

Finally, the reward function $\mathcal{R}$ scores molecules according to their alignment with desired chemical properties, which can involve complex material simulations. The underlying property computation of the reward function further informs the dynamics of the MDP imposed by text representation. For example, docking scores are used to estimate the binding affinity between ligands and protein targets. We discuss reward functions for molecules in more detail in section Section 5.

**Pretraining policies for molecule discovery.** To advance effectively within this vast search space, we make use of datasets containing a large number of drug-like molecules in text format (Irwin et al., 2012; Sterling and Irwin, 2015b; Mendez et al., 2019). This data is used to train an autoregressive model to predict tokens that conform to the grammar for drug-like molecules, instead of the random texts that are generated from a randomly initialized policy, thereby significantly simplifying the exploration problem. In particular, we pretrain a network $p_\phi$ on the self-supervised objective of next-token prediction. Although large language models can be trained with other objectives, such as corrupted text reconstruction (Edwards et al., 2022b), these models are not a good fit for our purposes since they cannot generate diverse and valid molecules without access to carefully designed prompts.

$$\min_\theta \mathbb{E}_{A \sim D} \left[ \sum_{t=1}^{H} - \log p_\theta(a_t = A_t \mid A_{t-1}, \cdots A_0) \right]. \tag{2}$$

In practice a minibatch of sequences $\{A^1, \cdots, A^m\}$ are sampled from the prior dataset $D$ to evaluate the loss function in Equation 2, and the parameters are trained using gradient descent.

## 4.1 RL FOR MOLECULE GENERATION

To generate a molecule, a ChemRLformer policy $\pi_\theta(a_t \mid s_t)$ is allowed to autoregressively sample tokens for a fixed number of timesteps $H$. The start state $s_0$ is always a beginning-of-sequence token [BOS], and the agent can terminate early by taking the end-of-sequence action [EOS]. Figure 1, shows how an RL policy can construct a Benzene molecule. Since we are only interested in the properties of the final molecule, there are no intermediate rewards and the goal of the RL policy is to maximize the expected scalar reward corresponding to the final constructed molecule, $r(s_T)$. Thus, assuming a discount rate $\gamma = 1$[1], Equation 1 can be rewritten more simply as:

$$\max_\theta \mathbb{E}_{s_T \sim \pi_\theta} \left[ r(s_T) \right] \tag{3}$$

where $s_T = [BOS][a_0][a_1] \cdots [EOS]$, is sampled autoregressively from the policy.

Our experiments use the policy gradient algorithm (Sutton et al., 1999b) to train the RL policy because it is known to achieve state-of-the-art performance amongst RL for molecular optimization (Olivecrona et al., 2017). Deep RL policies are able to learn the non-linear global structures of molecular texts which, as we show in section 5, enables them to generalize to novel and diverse molecules. However, training RL policies from scratch is time-consuming and can make the exploration problem infeasibly difficult. Next, we explain how we adapt recent language modelling techniques to pretrain the RL policy.

**RL fine-tuning** The pretrained model can directly be used to sample novel drug-like molecules. These molecules, however, are not optimized for any particular property. Note that given our definition of the state $s_t$ as the concatenated history of all previous actions, this pretrained network is exactly analogous to the policy network in Equation 1. Hence, by initializing $\pi_\theta = p_\phi$, and $\theta = \phi$, we can fine-tune this pretrained network by optimizing Equation 1 via the policy gradient algorithm - REINFORCE (Sutton et al., 1999a). We need only to define a reward function $r(s_T)$ which scores molecules according to their alignment with the desired properties. In the following experiments, we show that this fine-tuning is vital for ChemRLformer to sample better molecules. We also highlight the importance of pretraining and study how the size and quality of the prior data affect the downstream ability of RL to search for high-value molecules.

## 5 EXPERIMENTAL RESULTS

Our proposed algorithm ChemRLformer uses the best combinations of choices resulting from assessing the performance across three dimensions: (1) what pretraining factors are important to improve RL for molecular discovery (Section 5.2), (2) how the use of recent text-based molecule grammars facilitates downstream RL exploration (Section 5.3); and, lastly, (3) which specific algorithmic changes are necessary to improve RL performance (Section 5.4).

## 5.1 EXPERIMENTAL SETUP

**Tasks.** We evaluate ChemRLformer against five different **docking** targets (Alhossary et al., 2015) (fa7, parp1, 5ht1b, jak2, and braf) previously explored in the literature (Yang et al., 2021; Lee et al., 2023). The docking scores used to estimate the binding affinity between ligands and protein targets are a complex function of the global molecular structure and have been proposed as chemically relevant benchmarks for molecule design algorithms (Cieplinski et al., 2021; Tripp et al., 2022). In addition to the docking targets, we also evaluate on 22 pharmaceutically-relevant oracle functions (Huang et al., 2021; Gao et al., 2022b; Brown et al., 2019) (**pytdc** tasks), which include tasks such as optimizing proxies of bioactivity, similarity to target molecules, and combinations of multiple physiochemical drug properties.

**Evaluation metrics.** We design our evaluation procedure with the final goal of identifying the best candidates to test in a wet lab. To discover such high-value candidate molecules, we use sota simulators that assign rewards to molecules by performing complex docking simulations (Alhossary et al., 2015) or using proxy models and chemical rules (Huang et al., 2021). Previous works limit the

---

[1]Because we operate in a finite MDP, a discount factor of 1 gives us an unbiased estimate of the true objective.

number of molecules sampled during evaluation to around 3000 for **docking** tasks (García-Ortegón et al., 2022; Yang et al., 2021; Lee et al., 2023) and 10000 for **pytdc** tasks (Gao et al., 2022b; Brown et al., 2019) due to the computational cost associated with these reward simulators. We allow up to 25000 unique oracle calls and up to 40000 total oracle calls (allowing repeats). We argue this better reflects the lower cost and availability of computing resources relative to wet-lab resources. From all the sampled molecules, the average score of the top-$k$ ($k = 1, 10, 100$) molecules is used as a performance metric. These top groups are an estimate of the algorithm's ability to discover a group of top-quality candidates that could be given to a wet lab for thorough testing. We report **pytdc** scores on a normalized basis between zero and one by default. Next, we normalize all docking scores by dividing them by -20 in our experiments. Additionally, we report *diversity* (Div.), defined as the averaged internal distance of the top 100 molecules, and *redundancy* (Red.), defined as the total number of oracle calls that an agent makes for an already evaluated molecule.

**Pretraining.** We study how the quality and size of prior data affect the downstream RL performance of ChemRLformer by pretraining a GPT (Radford et al., 2018) style transformer model on five datasets of varying sizes and quality and using the pretrained model as an initialization for the RL agent's policy network. See Table 2 for the name, size, and description of all datasets used in our work. We also rank all datasets based on their quality on **docking** and **pytdc** tasks. We determine the quality of a dataset by the performance of molecules sampled from the model pretrained on that dataset. The quality of ChemRLformer's pretrained model is evaluated using the *top-100* molecules sampled by the pretrained model under the same evaluation setup in Appendix A.2. By default, these open-sourced datasets contain a large number of drug-like molecules in SMILES format. For our experiments, we also convert all datasets to the SELFIES format. Lastly, three different architectures are compared: **fully-connected (FC)**, **recurrent (RNN)** - a GRU and **transformer** - GPT style autoregressive model, and compare them on downstream RL tasks.

Table 2: **Description of molecular datasets used for pretraining:** Datasets are ranked according to procedure described in Section 5.1. Two datasets have the same rank if their average performance lies inside one standard error of the other. The datasets are drawn from a subset of the Zinc (Sterling and Irwin, 2015b; Irwin et al., 2022) and ChemBL (Gaulton et al., 2012) databases.

| Dataset | Size | Docking Rank | Pytdc Rank | Description |
|---------|------|--------------|------------|-------------|
| CHEMBL | 1.2 M | 1 | 1 | Manually curated database of bioactive molecules with drug-like properties (Gaulton et al., 2012). |
| ZINC 250K | 250 K | 2 | 2 | ZINC database molecules curated for their pharmaceutical relevance and popularity (Gao et al., 2022b). |
| ZINC 1M | 1 M | 3 | 3 | Random molecules from $\approx$ 1.5 billion |
| ZINC 10M | 10 M | 3 | 4 | molecules from the ZINC database (Sterling and Irwin, 2015a). |
| ZINC 100M | 100 M | 3 | 4 | ZINC 1M $\subset$ ZINC 10M $\subset$ ZINC100M. |

All of our experiments on **pytdc** tasks are run across 5 random seeds. Since docking simulations are expensive and time consuming, we run all **docking** experiments across 3 random seeds. Experiments with different seeds use the same pretrained model which is only pretrained once for every dataset. Additional details about the task rewards, evaluation metrics, and the pretraining datasets and models are discussed in Appendix A.1, A.2, and A.3 respectively.

## 5.2 How does prior data affect the final performance of ChemRLformer?

In this section, we pretrain the REINFORCE policy on datasets of varying size and quality from Table 2. Our datasets vary from small (250K) to very large (100M) sizes. Due to the parallelizability of training on larger datasets, we use the transformer policy architecture for all experiments in this section. In natural language processing (NLP), pretraining transformer models on large and diverse unlabelled datasets have been found to perform well on downstream tasks using few-shot labeled data (Brown et al., 2020b). Yet, our results in Figure 2b indicate that the quality of the prior dataset matters more than its size. We compares the scores of the molecules generated by the policy after pretraining and after RL training. Figure 2 shows that the molecules sampled after pretraining on the *ChEMBL* dataset achieves higher scores, and hence is more aligned with both the **pytdc** and the **docking** tasks. As a result, the RL agent pretrained on the *ChEMBL* dataset outperforms all other agents, including the ones trained on 100 times more data.

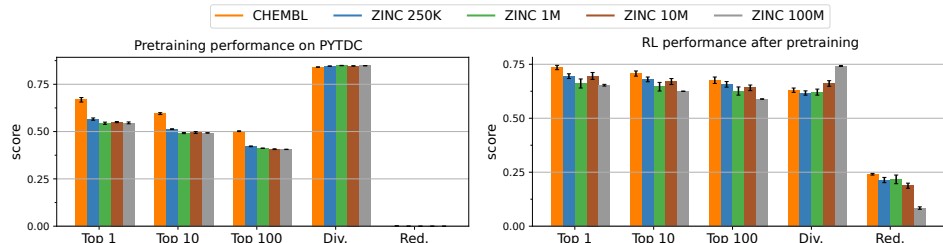

(a) Performance on SMILES-based molecular design with pretraining (left) and with pretraining and RL (right).

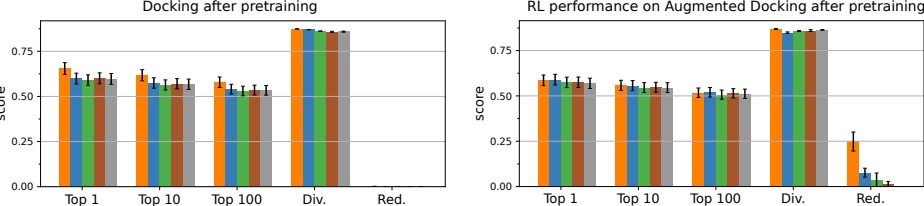

(b) Performance on SMILES-based molecular docking with pretraining (left) and with pretrainig and RL (right). Section 5.2 describes augmented docking setting with additional experiments shown in Appendix A.3. Div. and Red. are the diversity and redundancy scores described in section 5.1 respectively.

Figure 2: On the left pretrained performance on SMILES-based ChemRLformer. Higher-quality datasets, such as ChemBL lead to higher-performance for both **pytdc** and augmented docking. On the right is the performance after RL training. RL has a substantial benefit for **pytdc** tasks, while for docking tasks an *augmented docking* score is used to avoid reward hacking, see Figure 4 for details.

Results may seem surprising from an NLP perspective, but they make sense when viewed from an RL perspective. Pretraining using next token prediction Equation (2), is analogous to behavior cloning in this context, where the performance depends largely on the quality of the offline dataset (Ross et al., 2011; Ho and Ermon, 2016). These results suggest that ChemRLformer might benefit from better pretraining objectives, that go beyond simple imitation learning, when trained on large and diverse offline datasets (Kumar et al., 2023; Farebrother et al., 2023).

### 5.3 TEXT REPRESENTATIONS AND ARCHITECTURES FOR CHEMRLFORMER

Starting with a REINFORCE agent, we isolate the effect of various text representations for molecules and policy network architectures on performance. All experiments in this section use **ZINC-250k** dataset for pretraining. Similar results obtained for other datasets are shared in the following sections. Whenever we show normalized results across different experiments, we add the individual plots in Appendix C.1.

**Text representations.** In Figure 3 we compare ChemRLformer agents using different architectures and tasks across environments that base their dynamics on SELFIES and SMILES. The results show normalized scores across all architectures. Consistent with prior work (Gao et al., 2022b) we find that SMILES-based polices generally outperforms SELFIES-based policies. On all **pytdc** tasks and architectures, ChemRLformer agents based on SMILES consistently achieve better rewards when compared to SELFIES-based agents across all reward metrics. Although more subtle, we observe a similar theme in the **docking** tasks where SMILES achieves higher rewards than SELFIES on all top-K metrics. Another consistent theme in the results is that even though the diversity of top-100 molecules obtained by SELFIES is higher, the redundancy of SELFIES agents is higher as well. This means that SELFIES-based ChemRLformer agents explore a much smaller region of the molecular space. These results suggest that the rules which allow SELFIES strings to always be converted into a valid molecule can actually be detrimental to the agent's exploration and search for high-value molecules, more details in Appendix C.1.

**Architectures.** The results in Figure 4 show that the **transformer** and **RNN** have similar performance on all tasks. On the **pytdc** tasks, **FC** achieves worse performance than other

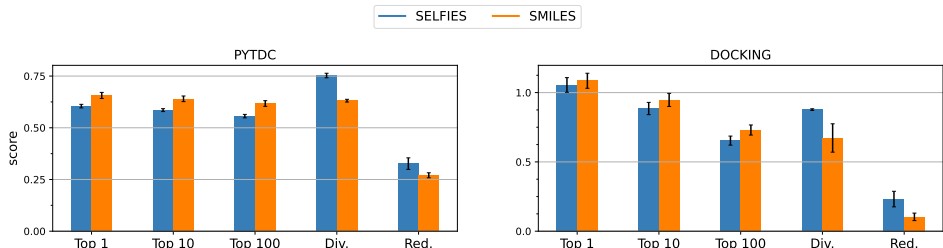

Figure 3: **Comparison between SELFIES and SMILES:** The SELFIES representation makes it relatively difficult for ChemRLformer agents to explore effectively leading to generally lower performance on pytdc and docking while scoring higher on diversity. Scores are reported for the transformer model and are averaged across all reward functions.

architectures specially made to handle strings, as expected. However, on **docking** tasks, **FC** obtains unusually high rewards. We find that this method performs a type of *reward function hacking* (Amodei et al., 2016; Skalse et al., 2022; Everitt, 2019) by exploiting a corner case of the docking-based reward function which provides high rewards for long strings of Carbon and Nitrogen atoms together. To evade the reward hacking of docking scores, we constructed an *augmented docking* score function with commonly used oracles (QED and SA scores) based on previous work (Lee et al., 2023) (See Appendix A.3 for more details). This finding shows that the REINFORCE agent can search the space well and, in this case, can be used to expose issues with the current design of reward functions.

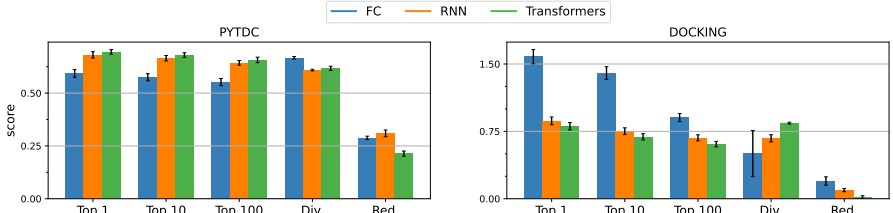

Figure 4: **Comparison of different policy architectures:** No single architecture clearly outperforms for molecular ChemRLformer. Although FC does better on the docking tasks, our analysis shows that it learns to exploit the docking function as opposed to designing high-value molecules. More details about ways to tackle this issue are given in Appendix C.3. Additional experiments for comparing transformers and RNNs are shown in Appendix C.5. These experiments use the smiles text representation.

## 5.4 REVISITING RL ALGORITHM DESIGN CHOICES FOR CHEMRLFORMER.

**Replay buffers and hill climbing.** In off-policy deep RL, a replay buffer is generally used to store and reuse previous trajectories for training. Although text-based RL algorithms are trained on-policy, prior work has proposed using a replay buffer to improve performance Blaschke et al. (2020b). Standard replay buffers Mnih et al. (2013) throw away the oldest trajectories as newer ones arrive. But many text-based RL algorithms propose to

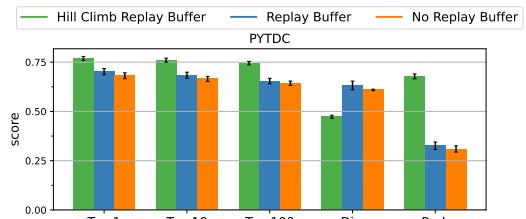

Figure 5: Hill climbing buffer lead to 13% improvement in Top-100 rewards.

use hill-climb replay buffers, that randomly sample a batch of molecules from the highest scoring molecules seen so far and add them to the current mini batch Thomas et al. (2022). In Figure 5, we see that using the hill-climb buffer results in a significant performance boost for ChemRLformer, whereas using a standard buffer does not contribute much. Notably, the use of a hill-climb replay buffer reduces diversity and increases redundancy quite substantially. The following two experiments involve combining regularisation terms with the RL objective in Equation 3. The coefficients on these extra terms can largely affect the final performance. To make a fair comparison,

we perform hyper-parameter tuning over six different values for every new regularisation term with details provided in Appendix B.

**Should the policy to stay close to the pretrained model?** Pretrained models carry information on how to build valid drug-like molecules. To ensure that ChemRLformer agents do not stray far away from the space of valid drug-like molecules during exploration, Olivecrona et al. (2017); Gao et al. (2022b) constrain the KL divergence between the policy and the pretrained model by adding a KL penalty to the policy gradient loss function in

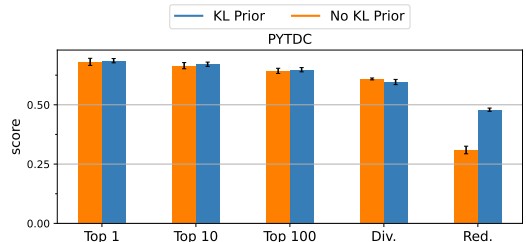

Figure 6: Our experiments show little difference in performance for multiple KL regularization terms.

Equation (3). Prior works show that adding this penalty helps the agent achieve better sample efficiency (Gao et al., 2022b). Yet, our results in Figure 6 suggest that, when you increase the number of oracle calls in simulation, adding this penalty does not yield any additional benefit while substantially increasing the GPU memory requirement, especially when using larger models. Since invalid molecules correspond to zero rewards, the ChemRLformer agent is able to learn to avoid invalid structures on its own merit.

**Regularizing the policy's likelihood for exploration.** RL agents classically face an exploration-exploitation dilemma, which can lead to agents getting stuck in sub-optimal local maxima when not well balanced. ChemRLformer agents are not immune to this dilemma. Upon encountering good, but sub-optimal molecules, an agent may adjust its policy to increase the likelihood of sampling these sub-optimal molecules and, without sufficient exploration, fail to discover higher-value regions of policy space. This can be particularly detrimental during the initial learning stages.

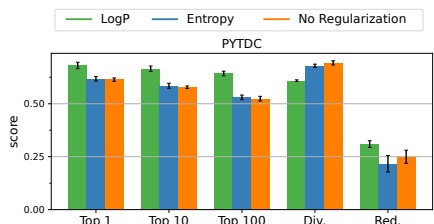

Figure 7: Different likelihood penalization for exploration. Log P regularization is a better choice for efficient exploration for ChemRLformer.

To combat this issue, entropy regularisation, which adds a $\log \pi(s)$ term to the RL loss, has been proposed (Haarnoja et al., 2018). This encourages the RL policy to explore states with lower likelihood values. Similarly (Olivecrona et al., 2017) adds a Log p regularizer, which penalizes higher likelihood values by adding a $-1/\log \pi(s)$ term to the RL loss. In Figure 7, our results show that although an entropy regularizer leads to lesser redundancy, the Log p regularizer boosts performance significantly by exploring more efficiently. The Log p regularizer only penalizes the agent for being extremely certain (likelihood $\xrightarrow{\text{tends to}} 1$) about its actions, and is mostly agnostic for lower likelihood values. This penalty is a much better choice for ChemRLformer as it only activates when stuck in a local optimum of molecular space.

## 6 CONCLUSION AND FUTURE WORK

We present ChemRLformer that resulted from our empirical study of multiple algorithmic components of text-based molecular design. For future practitioners, our method suggests the following philosophy: (1) Using SMILES is a better choice than SELFIES. (2) When collecting data for pretraining, the quality of molecules matter much more than the number of molecules. (3) Both transformer and RNN architectures achieve similar performance across all tasks using current datasets. (4) Incorporating components such as a hill-climb buffer and Log P regularization yields substantial performance improvements. Conversely, introducing KL regularization or opting for more intricate actor-critic algorithms may result in diminished performance, at the cost of more hyperparameters and memory resources. While our analysis focused on model-free RL algorithms, learning a reward model in a sample efficient manner is an exciting area of future work. Our analysis also shows that RL agents were able to *hack* the reward functions suggesting that there is space to improve on the metrics used for molecule quality.

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

## A  EXPERIMENTAL SETUP

In this section, we provide additional details about the tasks, evaluation metrics and pretraining models and data used in our work.

### A.1  TASKS

**Pydtc tasks.** These tasks are a set of 21 pharmaceutically-relevant oracle functions, which have been commonly used in prior work (Brown et al., 2019; Gao et al., 2022b; Huang et al., 2021) for evaluating performance across molecular discovery algorithms:

- QED: A quantitative estimate of drug-likeness calculated using a set of rules.
- DRD2, GSK3$\beta$, and JNK3: Classical machine learning models (SVMs and random forests) that provide an estimate of properties like target affinity or susceptibility towards a disorder.
- Celecoxib, Troglitazone, and Thiothixene rediscovery: An estimate of smiles text similarity, based on tanimoto metric, towards a target molecule.
- Albuterol and Mestranol similarity: Generate molecules similar to a target molecule.
- Isomers_c7h8n2o2 and isomers_c9h10n2o2pf2cl: Generate molecules corresponding to a target molecular formula.
- Median1 and Median2: Generate molecules that are maximally similar to several target molecules.
- Osimertinib_mpo, fexofenadine_mpo, ranolazine_mpo, perindopril_mpo, amlodipine_mpo, sitagliptin_mpo, zaleplon_mpo: Generate molecules that maximize multiple properties of a targeted drug.
- valsartan_smarts: Generate molecules that contain a certain SMARTS pattern and certain physicochemical properties.

Most of these tasks are from the GuacaMol benchmark (Brown et al., 2019). All oracles are calculated using the Python API provided by Therapeutics Data Commons (Huang et al., 2021) and more details for these tasks can be found on their website.

**Docking tasks.** We used QuickVina 2 (Alhossary et al., 2015) for calculating docking scores using the same default configuration parameters as prior works (Yang et al., 2021; Lee et al., 2023). For example, we used exhaustiveness $= 1$, and modes $= 10$. We choose 5 different protein targets to calculate docking scores: fa7 (FA7), parp1 (PARP-1), 5ht1b (5-HT1B), jak2 (JAK-2), and braf (BRAF). These targets were chosen by (Yang et al., 2021; Lee et al., 2023) because the docking simulators for these targets work fairly well when compared to the ground truth. In our experiments in Section 5.3, we found that text-based RL algorithms were easily able to produce chemically trivial molecules that have very high docking scores. To understand the complexity of computing docking scores, we report the time taken to dock 1000 molecules in parallel using 12 CPUs table 3. We also provide the time taken to run the RL algorithm on these 1000 molecules after their docking scores are available.

Table 3: **Time complexity of docking score evaluation** : More than half the running time is spent evaluating the docking scores.

| Number of molecules | Docking time | RL update time |
|---|---|---|
| 1000 | 130 seconds | 74 seconds |

**Augmented docking tasks.** In our results for the standard **docking** tasks (Figure 3 and Figure 4), we found that using the simulated docking scores as rewards did not lead to chemically relevant molecules. Text-based RL algorithms were able to exploit their state and action spaces to design chemically trivial molecules that have very high docking scores. To tackle this issue of undesirable reward hacking, we tried a reward function based on prior works (García-Ortegón et al., 2022; Lee et al., 2023) that combine objectives for drug-like, and synthesizable molecules with docking scores. We call tasks corresponding to this new reward function as **augmented docking** tasks. Concretely, we chose the same reward function from (Lee et al., 2023)

$$r(s) = -\text{DS}(s)/20 \times \text{QED}(s) \times (10 - \text{SA}(s))/9, \tag{4}$$

Where DS is the docking score, QED and SA are quantitative estimates of drug likeness and synthesizablity respectively.

## A.2 EVALUATION METRICS

Most of the metrics we use are described in detail in Section 5.1. Here, we provide additional details about the diversity metric. We calculate the diversity of the top 100 molecules sampled by the algorithm, where higher diversity is considered better given that it increases the chances for success in further wet lab experiments. In our experiments, we use the diversity evaluator from TDC (Huang et al., 2021), which defines the diversity of a set of molecules as the average pairwise Tanimoto similarity between Morgan fingerprints of the molecules. See Section 2 of (Benhenda, 2017) for exact details of how Tanimoto similarity is calculated.

## A.3 PRETRAINING

In this section, we provide more details about the pretraining datasets and models used in our experiments.

**Pretraining datasets.** The ZINC 250k dataset contains approximately 250k molecules from the ZINC database (Irwin et al., 2012), chosen for their pharmaceutical relevance, moderate size, and popularity (Gao et al., 2022b). The CHEMBL dataset (Mendez et al., 2019) consists of approximately 2M manually curated drug-like molecules. The other 3 datasets consist of randomly selected subsets of the ZINC-15 dataset (Sterling and Irwin, 2015b) that obey some chemically imposed mild constraints (Irwin et al., 2022). We test three subsets of different sizes: (1) ZINC 1M (2) ZINC 10M, and (3) ZINC 100M, to test the impact of scaling the size of pre-training data. These datasets and data-subsets, including their vocabularies, will be shared in an easily accessible format upon acceptance.

Removing outliers and unusual non drug-like compounds helps to keep the vocabulary small and improves the quality of the generative model (Blaschke et al., 2020a). To achieve this, we filter all datasets by removing molecules which contain 1) less than 10 or more than 50 heavy atoms and 2) molecules *other than* Carbon, Nitrogen, Oxygen, Fluorine, Silicon, Chlorine and Bromine. We also canonicalize and sanitize all molecules using RDKIT (Landrum et al., 2013). For experiments that apply SELFIES, we convert all datasets to SELFIES using the Python API provided by (Krenn et al., 2020) (Version: 2.1.1).

Apart from the experiments shown in the main paper, Appendix C.2 contains additional experiments comparing text-based RL agents across different pretraining datasets.

**Pretraining models.** In Table 4 we provide details about the pretraining modes which we use in our experiments. Upon acceptance, we will open-source our code and release the pretrained weights to support reproducible research.

We select network sizes that have been commonly used in RL (Blaschke et al., 2020a; Yarats et al., 2021). Although conducting a study of scaling the model size (Kaplan et al., 2020) is out of the scope of our work, we believe that it is a promising direction for future.

Since the fully connected model can only take fixed length inputs, we always input a molecular text padded to a certain maximum length (we used length 100 in our experiments). This padding is done

Table 4: **Description of model architectures used for pretraining**

| Model | Number of Parameters | Description |
|---|---|---|
| FC | $1.07 \times 10^7$ | FC is a fully connected neural network with 3 hidden layers of 1024 size each. |
| RNN | $4.17 \times 10^6$ | RNN is a recurrent network which consists of 3 GRU layers of hidden sizes 512 each. |
| TRANSFORMER | $4.78 \times 10^6$ | GPT (Brown et al., 2020b) style transformer with 6 layers, 16 heads and 256 embedding dimensions. |

using a special token [PAD] to convey that corresponding tokens should not be considered while deciding the value of the text.

**Pretraining experimental details.** We pretrain FC, RNN and transformer architectures on the ZINC 250K dataset and pretrain a transformer on all other datasets. All models are pretrained using the PyTorch (Paszke et al., 2019) framework. All models used an initial learning rate of $1e - 3$, with a cosine learning rate schedule (Loshchilov and Hutter, 2017). FC and RNNs used a batch size of 128 and were trained for 10 epochs. All transformers were trained for 5 epochs, with the largest batch size that we could fit in the memory of a single NVIDIA RTX A6000 GPU, for example, a batch size of 2048 for pretraining the transformer on ZINC 100M dataset. We made sure that all models were trained until convergence. On the ZINC 250K SMILES dataset, the FC, the RNN and the transformer model achieved a validation loss of 29.417, 22.507, and 22.923 respectively.

### A.4 RL FINETUNING

The pretrained model is further trained using the policy gradient algorithm, REINFORCE (Sutton et al., 1999b). Given the reward function $r(s_H)$ corresponding to the text $s_T$, this algorithm optimizes the loss function

$$\min_{\theta} J(\theta) = - \left[ \sum_{t=1}^{H} \log p_{\theta}(a_t = A_t \mid A_{t-1}, \cdots A_0) r(s_H = [A_0, \cdots A_H]) \right], \quad (5)$$

where $A_t$ is the token sampled by the agent at time-step $t$.

### A.5 REPLAY BUFFER

Although REINFORCE is an on policy method, we investigate the use of replay buffer to improve its sample efficieny Blaschke et al. (2020b). To do this, we store recent molecules sampled by the agent in a replay buffer. In our experiments we store the 100 most recent molecules. At every training update, we append a small batch of molecules sampled from the replay buffer to the on-policy batch of molecules. The entire new batch is used to update the policy using the policy gradient appendix A.4.

## B HYPERPARAMETER TUNING

We conduct a common hyperparameter tuning strategy for all experiments. Specifically, we conduct hyperparameter tuning for

- Learning rate for different architectures Figure 4 and text grammars Figure 3.
- Coefficients for different likelihood regularizations Figure 7.
- Coefficients for KL regularization loss term Figure 6.

We select three tasks from the **pytdc** tasks, i.e., troglitazone_rediscovery, sitagliptin_mpo, and median2 for hyperparameter tuning. For each hyperparameter, we select a set of 5 evenly spaced

realistic values and run 5 random seeds of RL experiments per hyperparameter value. We select the hyperparameter value that achieves the best average score of the top-100 molecules as the final value for running all the experiments. We report the hyperparamters used for the policy gradient training in table 5.

Table 5: **Hyperparamters**

| Name | Value |
|---|---|
| Maximum number of unique molecules | 25000 |
| Learning rate | $5.00 \times 10^{-4}$ RNN and FC
$1.00 \times 10^{-4}$ Transformer |
| Batch size | 64 |
| Log p coefficient | 5 |
| KL coefficient | $1.00 \times 10^{-3}$ |

## C  RESULTS

### C.1  TEXT REPRESENTATIONS AND ARCHITECTURES FOR RL

Here, we present additional results from subsection 5.3. Figure 8 shows that SMILES are a better molecular grammar when compared to SELFIES across all architectures, for the text based RL algorithms that we consider. Figure 9 compares various architectures, while keeping the molecular grammar fixed to SELFIES. The results in Figure 9 reflect our findings in Figure 4 that no single architecture clearly outperforms for molecular text-based RL. It also shows the reward hacking behavior of the **docking** tasks by the FC based RL agent.

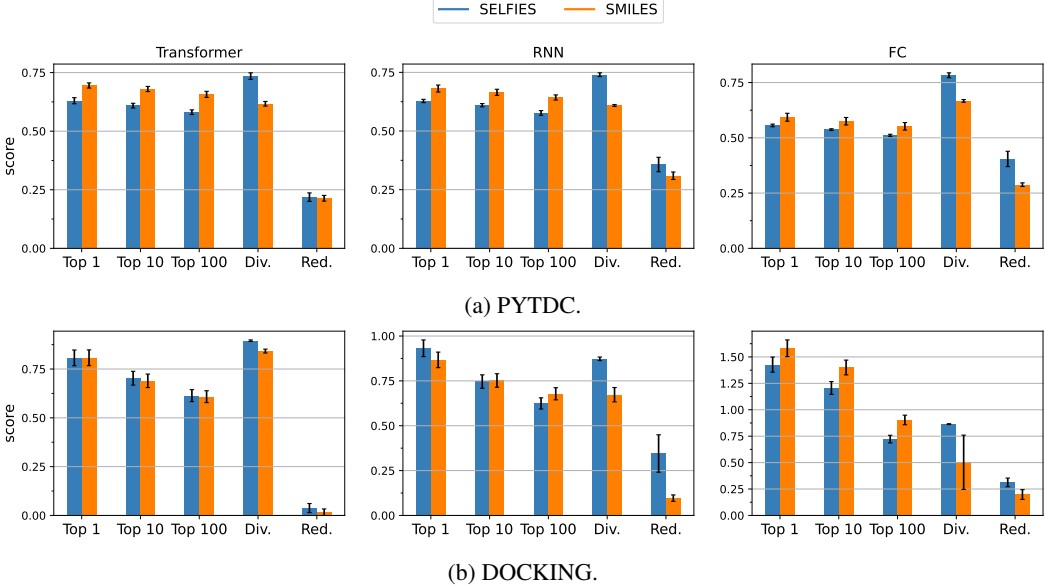

(a) PYTDC.

(b) DOCKING.

Figure 8: Comparison between SELFIES and SMILES across different architectures. These figures are the individual plots corresponding to the normalised plot show in Figure 3.

The reason for lower value molecules for SELFIES environments can be explained by the SELFIES grammar that induces a flat optimization landscape. Many SELFIES strings can correspond to the same molecule, and in fact, once an *invalid* action is taken, any subsequent sequence of tokens will be ignored on the resulting molecule. This makes exploration of new molecules difficult (Krenn et al., 2020; Gao et al., 2022b). On the other hand, the benefit of SELFIES over SMILES in

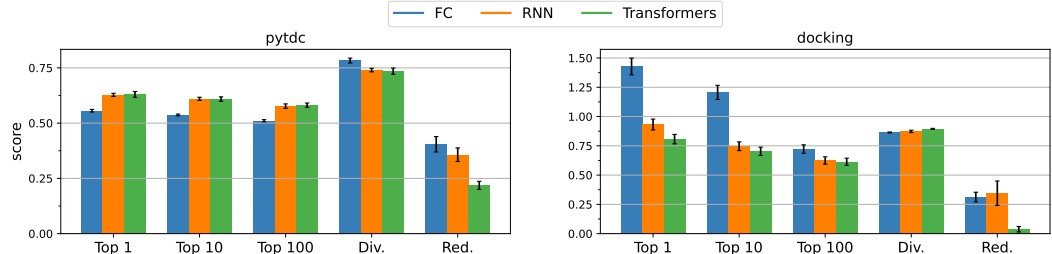

Figure 9: **Comparison of different policy architectures (SELFIES):** No single architecture clearly outperforms for molecular text-based RL. Although FC does better on the docking tasks, our analysis shows that it learns to exploit the docking function as opposed to designing high-value molecules.

eliminating invalid molecule generation is mitigated by our pretraining process, which initializes SMILES-based policies with a strong bias toward generating valid molecules. Overall, we find that SMILES-based policies, when combined with pretraining, are more effective at exploring and finding high-value molecules.

## C.2 PRETRAINING FOR RL

Figure 2 (right) shows the top docking scores obtained by RL agents pre-trained on different datasets when trained with on the **augmented docking** tasks. In Figure 10, we show the actual augmented rewards obtained by the RL agent. These results suggest that the augmented docking score is a complex reward function as the RL agent is achieved minimal improvement over the prior agent. To verify this hypothesis, we increased the molecule budget of the RL agent by 10 times. We indeed see that RL agents corresponding to all prior-datasets exhibit considerable improvement. Text-based RL algorithms learn to search more efficiently when provided with more compute.

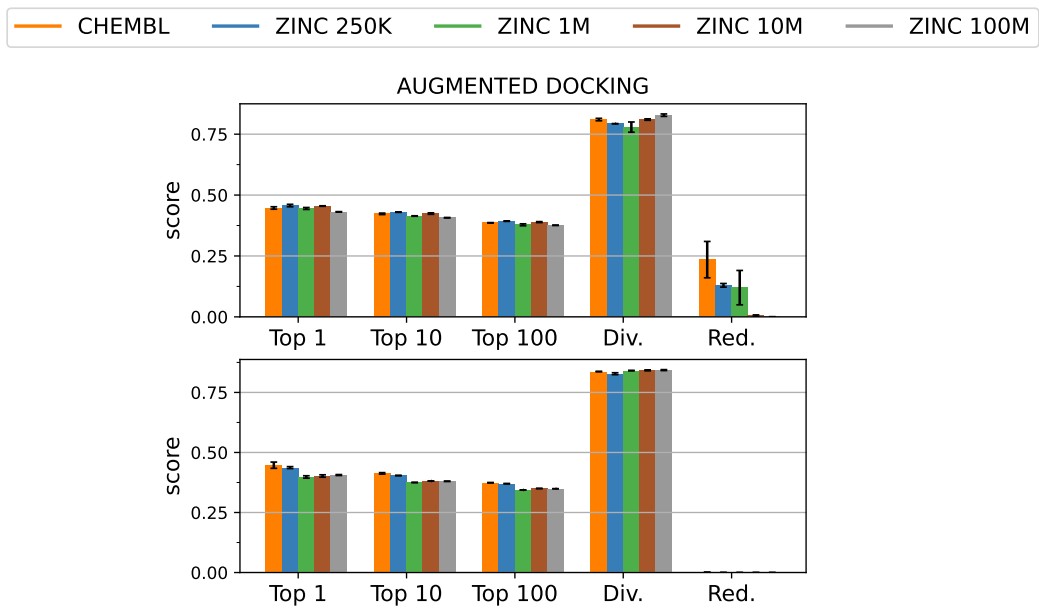

Figure 10: This figure shows the augmented rewards obtained by the RL agents (Top) and data quality (Bottom) of different datasets. See subsection A.1 for how the augmented reward is calculated.

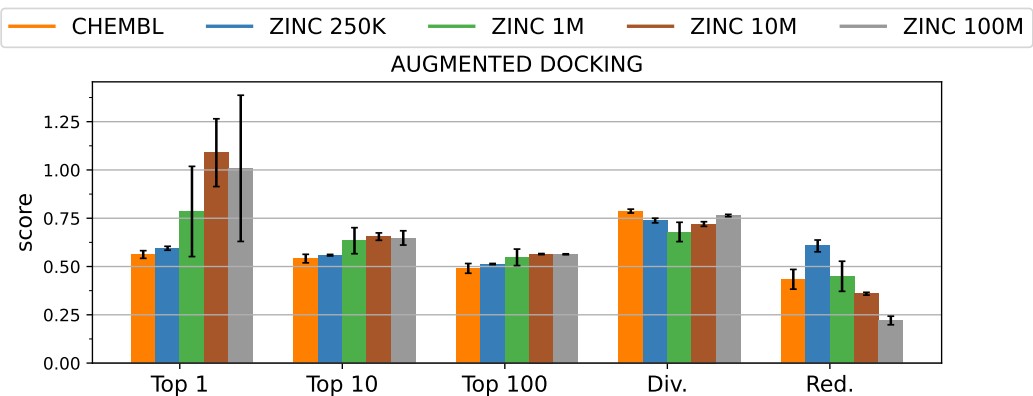

Figure 11: This figure shows the augmented rewards obtained by the RL agents trained for 10 times more molecules.

## C.3 REWARD HACKING

Figure 4 and Figure 9 show that text-based RL agents that are trained using fully connected neural networks are able to obtain unusually high rewards. This is probably because it is easier for FC agents to find actions that exploit the local structure of the reward function as RNNs and Transformers are inductively biased to find global solutions. This highlights an undesirable type of reward function hacking by the FC agent which provides high rewards for molecules with long strings of Carbon and Nitrogen atoms together. Similar to prior work (Lee et al., 2023), we augment the docking scores with objectives for drug-like and synthesizable molecules. See Appendix A for details of this task and Figure 2 and Figure 10 for results corresponding to this task. Our initial results on this task ( Figure 2 and Figure 10) suggested that the augmented reward function was more aligned towards chemically relevant molecules. We also noticed that the RL agents were not able to improve a lot over the prior baselines for this task. To verify whether the low performance of RL agents was because of less training data or the augmented reward function was indeed a more realistic and robust reward function, we repeated the experiments in Figure 10 with a ten times higher training budget. Given more data, all RL agents showed considerable improvements over the priors. This experiment also revealed that the agents pre-trained on ZINC 1M, ZINC 10M and ZINC 100M, were able to exploit the reward function to generate unrealistic yet highly rewarding molecules. These molecules have unusually low docking scores (less than -20). Our results highlight the need for an aligned and a more robust reward function to generate molecules for docking protein targets.

## C.4 ADDITIONAL RESULTS FOR THE IMPORTANCE OF ALGORITHMIC CHOICES FOR TEXT-BASED RL.

Section 5.4 compares various algorithmic components like replay buffers, hill climbing, KL regularisation towards the pretrained policy, and likelihood penalties and show results for **PYTDC** tasks. In this section, we repeat all the experiments from Section 5.4 on **augmented docking** tasks as well and reach the same conclusions. In Figure 15 we see that using the hill-climb buffer results in a significant performance boost, whereas using a standard buffer does not contribute much. Figure 16 shows that Log P regularization is a better choice for efficient exploration when it comes to text-based RL algorithms. In Figure 17 show that penalising the policy to move away from the pretrained policy does not improve performance.

## C.5 INSTABILITY OF TRANSFORMERS FOR ONLINE RL.

Many works (Parisotto et al., 2019) have pointed out the instability of training transformers using online reinforcement learning. To understand this in the context of text based RL, we compare a transformer and an RNN based agent on the augmented docking task. To probe whether pronounced effects of this instability are seen, we train both agents for 10 times more molecules (250K

Table 6: Performance of ChemRLformer across all PyTDC tasks.

| Target | Avg Top1 | Avg Top10 | Avg Top100 | Diversity_top100 | Redundant_count |
|---|---|---|---|---|---|
| drd2 | 1.000 (0.000) | 1.000 (0.000) | 1.000 (0.000) | 0.465 (0.052) | 29907.0 (1689.767) |
| qed | 0.948 (0.000) | 0.948 (0.000) | 0.948 (0.000) | 0.639 (0.064) | 26802.6 (460.301) |
| jnk3 | 0.896 (0.096) | 0.892 (0.095) | 0.883 (0.095) | 0.348 (0.094) | 29844.8 (3249.365) |
| gsk3b | 1.000 (0.000) | 0.997 (0.004) | 0.992 (0.006) | 0.407 (0.030) | 26923.4 (2663.734) |
| celecoxib_rediscovery | 1.000 (0.000) | 0.889 (0.007) | 0.847 (0.006) | 0.285 (0.031) | 28856.6 (1903.692) |
| troglitazone_rediscovery | 0.654 (0.098) | 0.643 (0.089) | 0.627 (0.082) | 0.364 (0.035) | 27425.4 (891.261) |
| thiothixene_rediscovery | 0.669 (0.005) | 0.669 (0.005) | 0.649 (0.006) | 0.403 (0.012) | 29158.4 (1492.667) |
| albuterol_similarity | 1.000 (0.000) | 1.000 (0.000) | 1.000 (0.000) | 0.529 (0.034) | 31551.2 (1310.567) |
| mestranol_similarity | 0.883 (0.124) | 0.873 (0.122) | 0.825 (0.109) | 0.371 (0.029) | 30395.2 (1516.911) |
| isomers_c7h8n2o2 | 1.000 (0.000) | 1.000 (0.000) | 0.950 (0.031) | 0.781 (0.056) | 33245.4 (1393.527) |
| isomers_c9h10n2o2pf2cl | 0.931 (0.007) | 0.925 (0.009) | 0.904 (0.014) | 0.598 (0.073) | 31457.4 (1893.718) |
| median1 | 0.415 (0.030) | 0.395 (0.027) | 0.379 (0.019) | 0.425 (0.050) | 27805.4 (1172.183) |
| median2 | 0.353 (0.033) | 0.347 (0.031) | 0.337 (0.027) | 0.323 (0.045) | 25513.0 (923.048) |
| osimertinib_mpo | 0.911 (0.005) | 0.907 (0.005) | 0.903 (0.005) | 0.389 (0.033) | 26083.6 (1287.568) |
| fexofenadine_mpo | 0.940 (0.020) | 0.938 (0.021) | 0.927 (0.019) | 0.479 (0.034) | 26143.0 (1100.390) |
| ranolazine_mpo | 0.879 (0.011) | 0.876 (0.010) | 0.872 (0.009) | 0.351 (0.050) | 23383.8 (1070.175) |
| perindopril_mpo | 0.637 (0.020) | 0.635 (0.020) | 0.632 (0.018) | 0.410 (0.067) | 26441.8 (938.579) |
| amlodipine_mpo | 0.827 (0.079) | 0.827 (0.080) | 0.817 (0.078) | 0.340 (0.066) | 28469.4 (1439.003) |
| sitagliptin_mpo | 0.573 (0.056) | 0.566 (0.056) | 0.547 (0.063) | 0.611 (0.025) | 30156.6 (1903.446) |
| zaleplon_mpo | 0.631 (0.019) | 0.623 (0.021) | 0.605 (0.021) | 0.542 (0.065) | 29838.2 (531.040) |
| valsartan_smarts | 0.000 (0.000) | 0.000 (0.000) | 0.000 (0.000) | 0.880 (0.002) | 0.0 (0.000) |
| normalised_score | 0.769 (0.010) | 0.760 (0.010) | 0.745 (0.009) | 0.473 (0.008) | 27114.3 (495.515) |

Table 7: Performance of just the pretrained model across all PyTDC tasks

| Target | Avg Top1 | Avg Top10 | Avg Top100 | Diversity_top100 | Redundant_count |
|---|---|---|---|---|---|
| drd2 | 0.962 (0.022) | 0.882 (0.018) | 0.484 (0.011) | 0.853 (0.002) | 9.600 (4.224) |
| qed | 0.948 (0.000) | 0.947 (0.000) | 0.944 (0.000) | 0.853 (0.005) | 9.600 (4.224) |
| jnk3 | 0.446 (0.153) | 0.327 (0.034) | 0.197 (0.004) | 0.868 (0.004) | 9.600 (4.224) |
| gsk3b | 0.748 (0.061) | 0.580 (0.027) | 0.363 (0.011) | 0.868 (0.002) | 9.600 (4.224) |
| celecoxib_rediscovery | 0.444 (0.019) | 0.399 (0.007) | 0.342 (0.002) | 0.829 (0.003) | 9.600 (4.224) |
| troglitazone_rediscovery | 0.304 (0.016) | 0.281 (0.004) | 0.250 (0.001) | 0.829 (0.002) | 9.600 (4.224) |
| thiothixene_rediscovery | 0.382 (0.013) | 0.353 (0.005) | 0.310 (0.001) | 0.806 (0.003) | 9.600 (4.224) |
| albuterol_similarity | 0.605 (0.022) | 0.561 (0.012) | 0.487 (0.005) | 0.847 (0.003) | 9.600 (4.224) |
| mestranol_similarity | 0.486 (0.015) | 0.448 (0.007) | 0.388 (0.003) | 0.833 (0.004) | 9.600 (4.224) |
| isomers_c7h8n2o2 | 0.948 (0.045) | 0.839 (0.032) | 0.556 (0.032) | 0.900 (0.003) | 9.600 (4.224) |
| isomers_c9h10n2o2pf2cl | 0.824 (0.029) | 0.740 (0.019) | 0.597 (0.011) | 0.880 (0.003) | 9.600 (4.224) |
| median1 | 0.285 (0.021) | 0.240 (0.005) | 0.198 (0.002) | 0.852 (0.003) | 9.600 (4.224) |
| median2 | 0.243 (0.013) | 0.215 (0.004) | 0.189 (0.001) | 0.822 (0.002) | 9.600 (4.224) |
| osimertinib_mpo | 0.812 (0.004) | 0.799 (0.002) | 0.771 (0.001) | 0.843 (0.001) | 9.600 (4.224) |
| fexofenadine_mpo | 0.723 (0.013) | 0.702 (0.004) | 0.670 (0.002) | 0.838 (0.003) | 9.600 (4.224) |
| ranolazine_mpo | 0.636 (0.020) | 0.551 (0.012) | 0.441 (0.010) | 0.856 (0.003) | 9.600 (4.224) |
| perindopril_mpo | 0.507 (0.021) | 0.472 (0.009) | 0.437 (0.003) | 0.811 (0.002) | 9.600 (4.224) |
| amlodipine_mpo | 0.629 (0.020) | 0.580 (0.011) | 0.521 (0.003) | 0.814 (0.002) | 9.600 (4.224) |
| sitagliptin_mpo | 0.438 (0.027) | 0.378 (0.013) | 0.284 (0.005) | 0.858 (0.004) | 9.600 (4.224) |
| zaleplon_mpo | 0.513 (0.013) | 0.484 (0.005) | 0.443 (0.003) | 0.830 (0.002) | 9.600 (4.224) |
| valsartan_smarts | 0.003 (0.007) | 0.000 (0.001) | 0.000 (0.000) | 0.876 (0.004) | 9.600 (4.224) |
| normalised_score | 0.566 (0.006) | 0.513 (0.002) | 0.422 (0.002) | 0.846 (0.001) | 9.600 (1.282) |

Table 8: Performance of ChemRLformer across all docking tasks.

| Target | Avg Top1 | Avg Top10 | Avg Top100 | Diversity_top100 | Redundant_count |
|---|---|---|---|---|---|
| 5ht1b | 17.467 (4.742) | 14.903 (2.295) | 13.131 (1.374) | 0.870 (0.019) | 16221.333 (11267.121) |
| jak2 | 13.200 (1.158) | 12.370 (0.790) | 11.294 (0.700) | 0.853 (0.031) | 19294.333 (10886.657) |
| fa7 | 18.933 (5.949) | 13.860 (1.549) | 11.349 (0.699) | 0.884 (0.017) | 5334.000 (6029.172) |
| braf | 21.700 (11.950) | 16.817 (5.709) | 12.366 (0.746) | 0.888 (0.013) | 13023.333 (6672.632) |
| parp1 | 21.867 (8.745) | 16.680 (2.110) | 14.310 (0.466) | 0.869 (0.024) | 15175.667 (10328.254) |
| normalised_score | 18.633 (2.823) | 14.926 (1.121) | 12.490 (0.397) | 0.873 (0.010) | 13809.733 (4172.842) |

Table 9: Performance of the just the the pretrained model across all docking tasks.

| Target | Avg Top1 | Avg Top10 | Avg Top100 | Diversity_top100 | Redundant_count |
|---|---|---|---|---|---|
| 5ht1b | 12.700 (0.283) | 12.250 (0.136) | 11.557 (0.083) | 0.866 (0.003) | 11.333 (1.886) |
| jak2 | 11.867 (0.287) | 11.193 (0.115) | 10.510 (0.077) | 0.874 (0.004) | 11.000 (1.414) |
| fa7 | 10.167 (0.125) | 9.837 (0.019) | 9.241 (0.018) | 0.873 (0.003) | 11.667 (3.300) |
| braf | 12.333 (0.125) | 11.853 (0.047) | 11.160 (0.091) | 0.869 (0.004) | 11.667 (1.247) |
| parp1 | 12.833 (0.047) | 12.323 (0.042) | 11.588 (0.039) | 0.869 (0.001) | 11.333 (4.028) |
| normalised_score | 11.980 (0.082) | 11.491 (0.013) | 10.811 (0.019) | 0.870 (0.001) | 11.400 (2.142) |

molecules). In figure Figure 12, we see that both agents perform comparably across all docking targets.

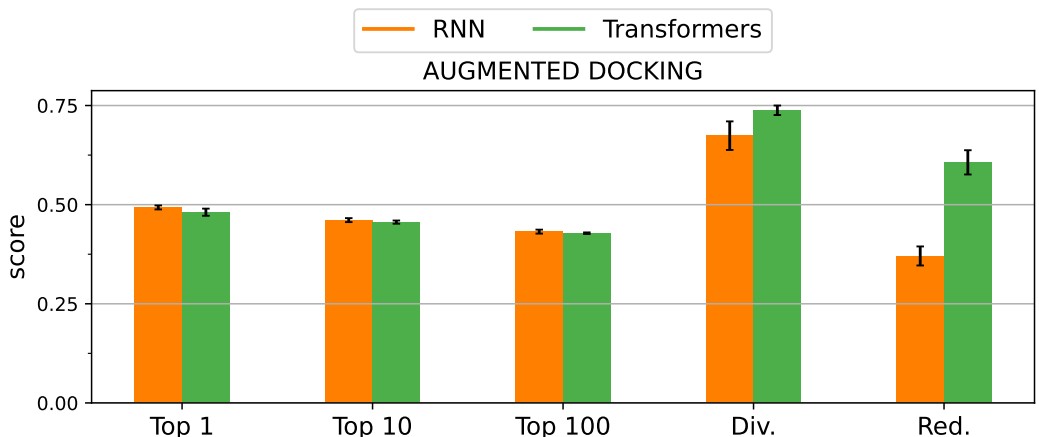

Figure 12: A transformer and an RNN based RL agent trained for 10 times more molecules on augmented docking scores.

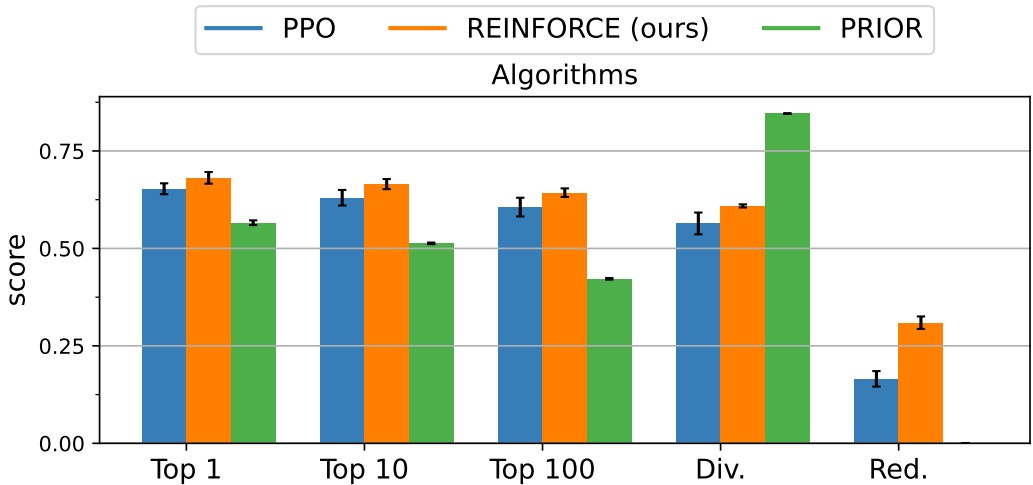

Figure 13: **Comparison with another RL algorithm PPO.** To keep the evaluation fair, we compared both the algorithms without the replay buffer and without the KL penalty. Both algorithms used logp regularization. We made sure that both algorithms use the same pretrained policy. On the molecular optimization tasks of PyTDC, our results indicate that vanilla policy gradient algorithms are more stable than actor critic algorithms like PPO and achieve higher performance. We believe this occurs because PPO learns a value function which is a difficult task when the reward function is sparse. This is indeed the case for molecular optimization. The agent only gets non zero rewards at the last step of the episode, when the molecule is scored. This makes the value function learning highly biased. This results resonates with the findings of previous work in molecular optimization (Cieplinski et al., 2021; Gao et al., 2022b).

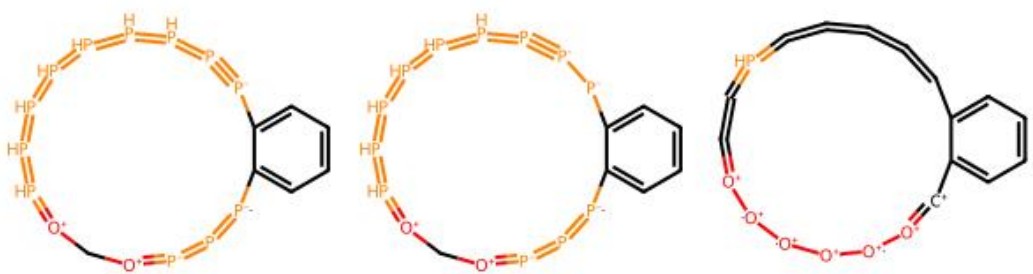

Figure 14: Some ChemRLformer agents are able to obtain unusually high docking scores by stacking together long chains and rings of sulphur, phosphorus or carbon atoms.

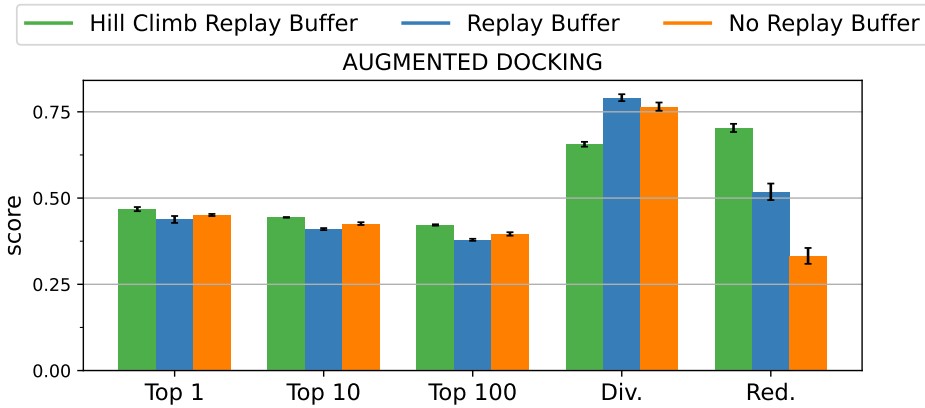

Figure 15: **Do replay buffers help?**

## D  DISCUSSION WITH DOMAIN EXPERTS.

Domain experts find the versatility of ChemRLformer in solving multiple tasks, especially expensive tasks like docking, useful for early-stage drug discovery. The reward-hacking insights are particularly useful as they show the shortcomings of docking score evaluations and the need for better evaluation, both in simulation and real-world experiments. We will add a deeper discussion on domain expert and application perspectives in the camera-ready version.

## E  REINFORCEMENT LEARNING FROM HUMAN FEEDBACK.

In recent years, Reinforcement Learning from Human Feedback (RLHF) has become a popular framework to train large language models to optimise a reward model. This reward model is meant to represent human preferences over outcomes. The reward model is trained using a datasets of sequence label pairs. These labels are provided by humans, depending on how *good* the sequence is. This recipe has been successful to finetune large language models Ouyang et al. (2022); Gao et al. (2022a).

While RLHF is a promising direction of future work, where chemists could interact and give feedback to the RL agent to generate desirable molecules, one reason why this will be more difficult in the space of molecular optimisation than natural language is that the expertise in chemistry is much more rare. And given that training a reward model requires large amount of labels, the current RLHF framework may not scale well for this use case. Right now, these experts are indirectly performing RLHF by creating simulations (e.g. Autodock) which we use in our experiments. Even with these simulations, which could be used to label unlimited molecules, we should focus on sample

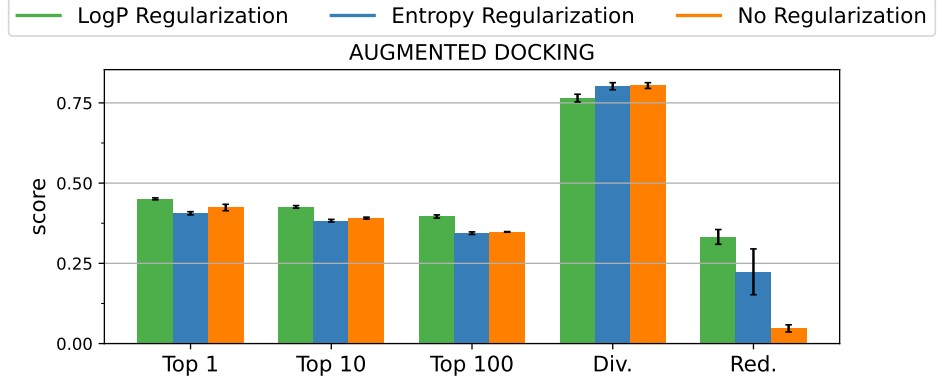

Figure 16: **Comparison of different likelihood penalization for efficient exploration**

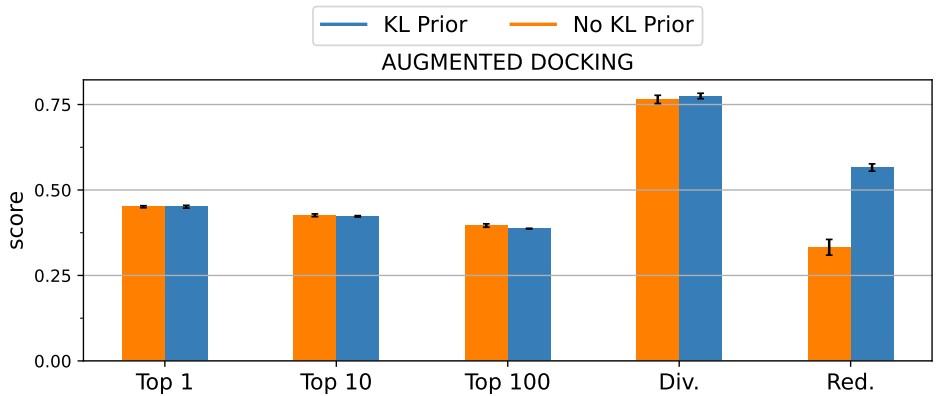

Figure 17: **Is KL regularisation with a prior necessary?**

efficient RL algorithms. The reason being that the simulators aren't perfect appendix C.3 and real chemists will only be able to give feedback on a limited number of molecules.

# F  RESULTS

In this section, we add images of the top 5 molecules found by ChemRLformer on each of the PyTDC tasks:

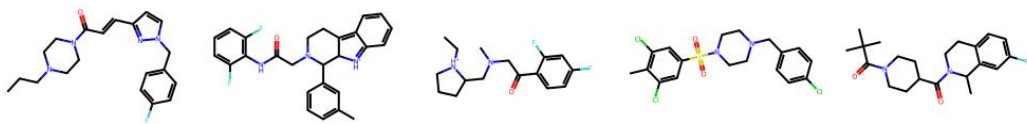

Figure 18: drd2.

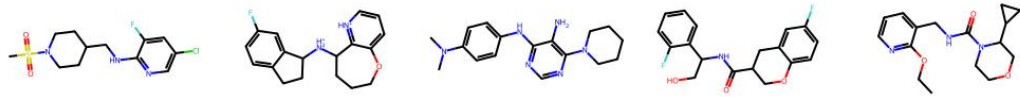

Figure 19: qed.

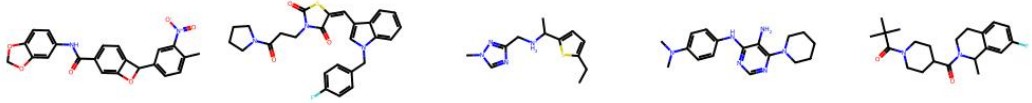

Figure 20: jnk3.

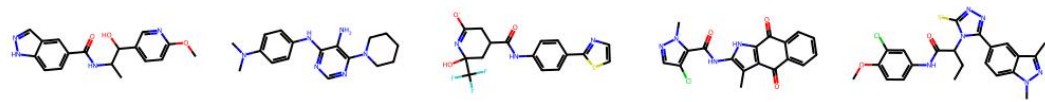

Figure 21: gsk3b.

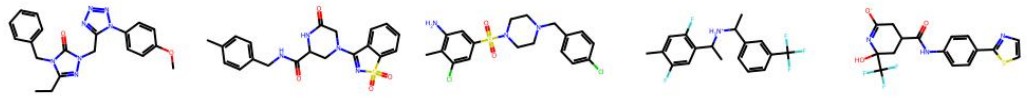

Figure 22: celecoxib rediscovery.

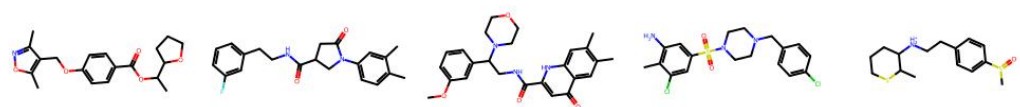

Figure 23: troglitazone rediscovery.

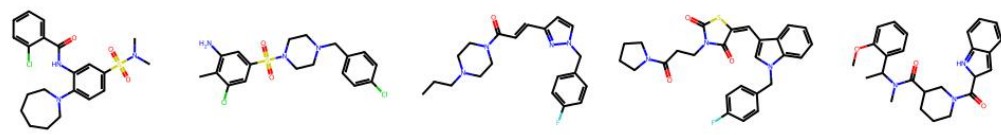

Figure 24: thiothixene rediscovery.

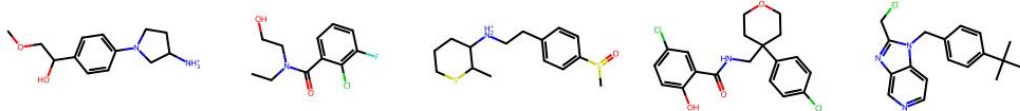

Figure 25: albuterol similarity.

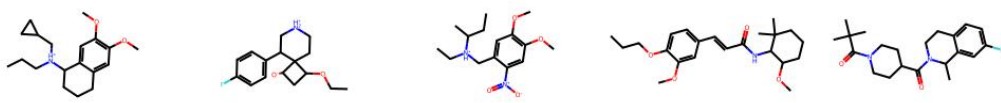

Figure 26: mestranol similarity.

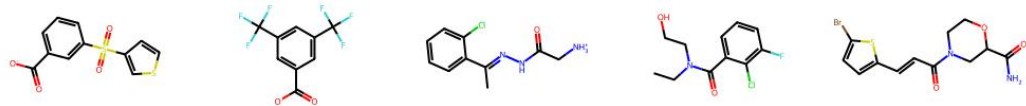

Figure 27: isomers c7h8n2o2.

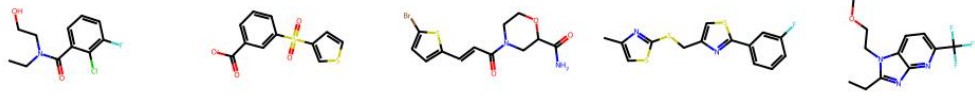

Figure 28: isomers c9h10n2o2pf2cl.

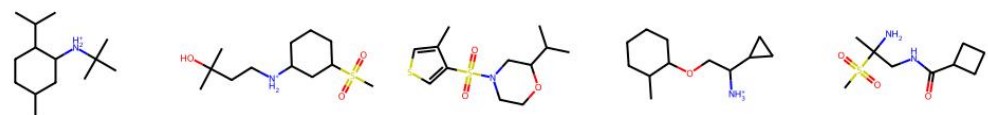

Figure 29: median1.

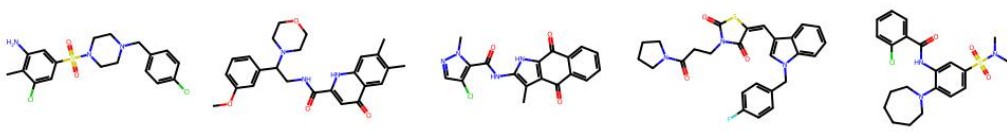

Figure 30: median2.

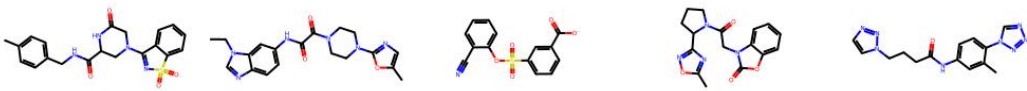

Figure 31: osimertinib mpo.

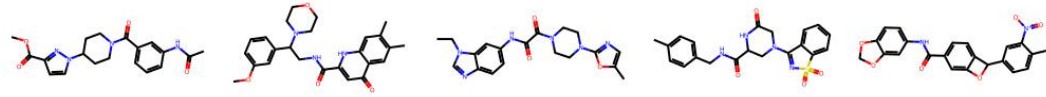

Figure 32: fexofenadine mpo.

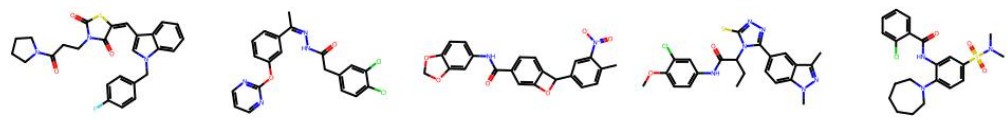

Figure 33: ranolazine mpo.

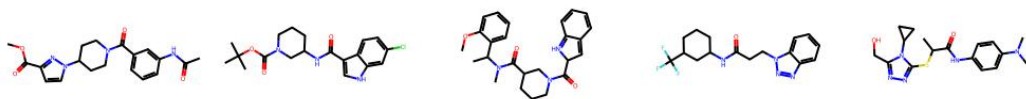

Figure 34: perindopril mpo.

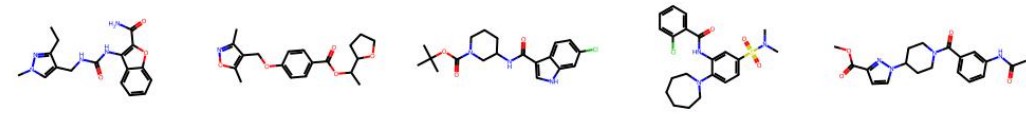

Figure 35: amlodipine mpo.

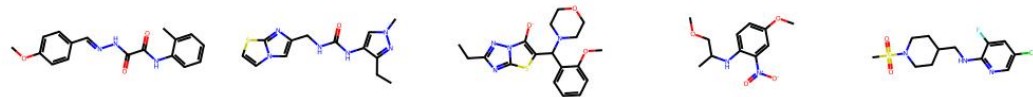

Figure 36: sitagliptin mpo.

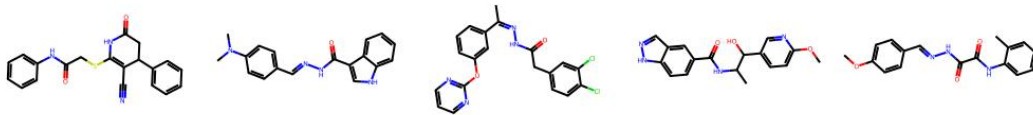

Figure 37:  zaleplon mpo.

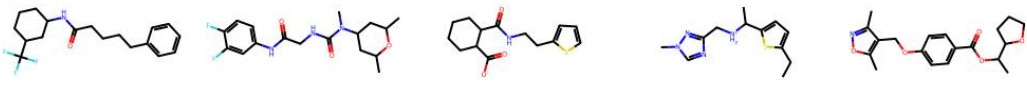

Figure 38:  valsartan smarts.

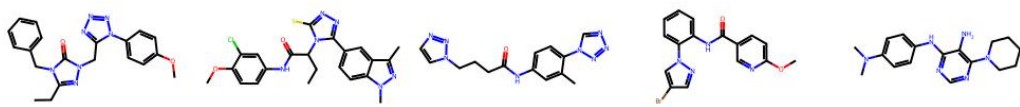

Figure 39:  deco hop.

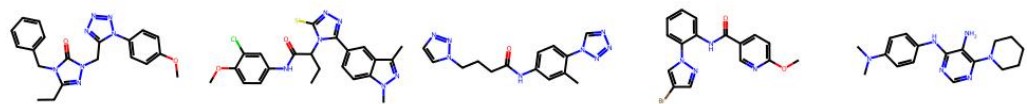

Figure 40:  scaffold hop.

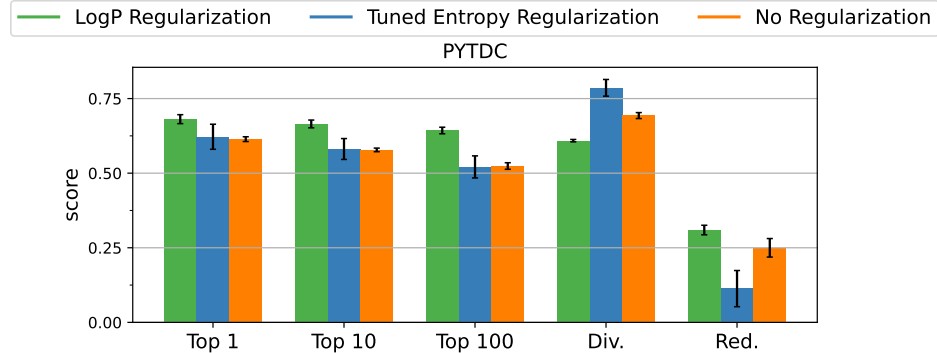

Figure 41:   We replicate the comparison in fig. 7 with an automatically tuned coefficient for the entropy Haarnoja et al. (2018). We follow Christodoulou (2019) to set the target entropy as $-0.98 \log(1/|A|)$, where $|A|$ is equal to the number of tokens in the vocabulary. Adding this feature does not improve the performance of entropy regularisation and as explained in section 5.4, Log P regularisation performs much better.

