# OpenReview forum: "Searching for High-Value Molecules Using Reinforcement Learning and Transformers"
_ICLR.cc/2024/Conference — ICLR 2024 poster_

### Official Review · Reviewer_feem · 2023-10-21

**Soundness:** 4 excellent
**Presentation:** 4 excellent
**Contribution:** 3 good
**Rating:** 8
**Confidence:** 4

**Summary:**

The paper presents a study of different design choices (problem representation, neural architecture, pretraining, etc) in the space of string-based RL for molecular discovery. The paper presents extensive experiments, including more than 25 molecule design tasks, to compare different design choices. The best combination of design choices is then used to propose a new RL-based molecular design algorithm (ChemRLformer). The paper also provides a thorough discussion of the results and provides valuable insights and recommendations for practitioners in the field of molecular discovery.

The paper is well written and organized. It is easy to follow and understand. The paper is also well motivated and provides a good introduction to the field of string-based RL for molecular discovery. I recommend the paper for acceptance.

**Strengths:**

- Technical Soundness: The paper demonstrates a high level of technical rigor. It conducts extensive experiments that compare various design choices (such as problem representation, neural architecture, pretraining, etc.) within the realm of string-based reinforcement learning for molecular discovery. Moreover, the paper offers a comprehensive discussion of the results and imparts valuable insights and recommendations for future research.

- Clarity and Organization: The paper is well-written and organized, making it easy to understand and follow. It effectively motivates the reader and provides a solid introduction to the field of string-based reinforcement learning for molecular discovery.

- Significance: The paper makes a substantial contribution to the field of molecular discovery. It offers a comprehensive comparative study of diverse design options (problem representation, neural architecture, pretraining, etc.) within the domain of string-based reinforcement learning for molecular discovery.

- Table 1 provides a nice summary of the different algorithms for text-based molecule design.

**Weaknesses:**

- Originality: While the paper offers a thorough numerical comparison of different design choices (problem representation, neural architecture, pretraining, etc.) in the context of string-based reinforcement learning for molecular discovery, its originality is somewhat constrained.

**Questions:**

- (Introduction) "...our own algorithm (MoLRL)" - what is MoLRL?
- (Table 1) What is the difference between "MoLRL" and "ChemRLformer (Ours)"? Is MolRL the general framework and ChemRLformer the best combination of design choices based on the ablation study? Please clarify.
- (Section 4) "The molecular design space is complex but the benefit from finding improved options is great." This sentence reads a bit too colloquially. Please rephrase.
- (Section 4) "However, the corresponding transition function induced in the graph representation of molecules is more complex as it is determined by the encoding/decoding rules of the chosen text representation." Are these constraints hard-coded in situ on the transitions dynamics or are they learned? Please clarify.
- (Figure 2) "(a) Performance on SMILES-based molecular design with pertaining (left) and with pretrianing and RL (right)." Please correct the typos "pertaining" and "pretrianing".
- (Figure 2) "b) performance on SMILES-based molecular docking with pertaining (right) and with pretrainig and RL (right)." First (right) should be (left). Please correct.
- (Figure 2) How does the "pretraining only" results (left column) are computed? Is this a zero-shot evaluation? I can't find the answer in the text. Please clarify.

---

> ### Author Response · Authors · 2023-11-17
> **Response to feem**
>
> Dear reviewer,
>
> We thank the reviewer for their feedback and summary of our work. We appreciate the reviewer’s assessment that the paper makes a significant contribution to molecular discovery and is technically sound and well-written. It seems that the main concern of the reviewer is the originality of the proposed algorithm. We agree that the basic components algorithms and design choices that we experimentally study have been used separately in prior work, ours is the first fair and extensive comparison at a large scale for string-based RL. To highlight prior work, we have included Table 1 in our paper, which contains relevant prior works that have used some of the methods that we incorporate.
>
> Below, we will answer your questions in more detail, and incorporate your feedback by making updates to the paper.
>
> > (Introduction) "...our own algorithm (MoLRL)" - what is MoLRL?
>
> This is a typo. We have substitute MoLRL -> ChemRLformer. Thank you for pointing this out.
>
> > Is MolRL the general framework and ChemRLformer
>
> This is a typo as well, they are the same method. We have fixed this in our updated paper.
>
> > This sentence reads a bit too colloquially.
>
> We have changed this sentence to : “The space of drug-like molecules is vast, and reinforcement learning methods hold great promise in improving the speed and reducing the cost of drug discovery.”
>
> > However, the corresponding transition function induced in the graph representation of molecules is more complex as it is determined by the encoding/decoding rules of the chosen text representation
>
> These transition dynamics are determined by the encoding / decoding rules of the chosen text representation. But the underlying constraints occur due to valencies and other chemical constraints that come into play when combining atoms and bonds to form molecules.
>
> > Please correct the typos "pertaining" and "pretrianing".
>
> We have corrected these typos in the updates paper.
>
> >  Please correct.
>
> We have corrected this in the updated paper, thank you.
>
> > How does the "pretraining only" results (left column) are computed? Is this a zero-shot evaluation?
>
> The pretraining only scores are calculated by sampling molecules from the pretrained model, without any RL training. We have added a sentence to clarify this in the updated paper (section 5.2, 1st para) : “We compare the scores of the molecules generated by the policy after pretraining and after RL training.”

---

> > ### Author Response · Authors · 2023-11-20
> > **Rebuttal Follow-up**
> >
> > Dear Reviewer, We hope that you've had a chance to read our responses and clarification. As the end of the discussion period is approaching, we would greatly appreciate it if you could confirm that our updates have addressed your concerns.

---

> > > ### Comment · Reviewer_feem · 2023-11-20
> > >
> > > I acknowledge that I have read the answers from the authors and the other reviewers. I thank the authors for their answers. I maintain that the paper is a valuable contribution that I recommend for acceptance.

---

### Official Review · Reviewer_AgdE · 2023-10-24

**Soundness:** 4 excellent
**Presentation:** 3 good
**Contribution:** 3 good
**Rating:** 6
**Confidence:** 4

**Summary:**

Authors propose Transformer-based approach with Reinforcement Learning tuning for molecular generation. Approach is tested using different datasets and wide ablation study is conducted.

**Strengths:**

Promising approach for the molecular generation. Experiments are conducted using three different datasets and impact of each of them is shown.

Good ablation study includes the design choices, different models architectures and even other RL algorithms beside REINFORCE are tested.

**Weaknesses:**

**Relevant work**

Some relevant works seem to be missing:

* MolGPT (https://www.ncbi.nlm.nih.gov/pmc/articles/PMC10232454/) also uses Transformer for molecular generation but without RL.

* Taiga (https://pubs.acs.org/doi/10.1021/acs.jcim.1c00600) uses Transformer and REINFORCE algorithm for the molecular generation with different properties optimization.

* Please also consider citing the relevant parallel work which applies offline RL for the similar problem: https://www.biorxiv.org/content/10.1101/2023.11.29.569328v1 (once it is public)

The second work seems to be very similar in terms of approach and I'm not sure about the novelty of the approach itself.

** Results presentation **

* I haven't noticed the comparison against any other methods.

* From the plots I couldn't understand what "Div." and "Red." columns stand for.

* I couldn't find reported scores for different tasks (e.g., similarity, QED or SA). It would be nice to see them.

* There are no samples of generated molecules. RL could hack reward functions and generate molecules which make no sense. Reward hacking is mentioned by the authors but including resulting molecules is essential.

**Questions:**

Please include relevant work which is mentioned in the **Weaknesses**

* What are the principal differences between you approach and Taiga?

* Can you compare against MolGPT, Taiga and REINVENT (or similar approach)?

* What are "Div." and "Red." columns?

* What are the scores for different reward components (e.g. QED)?

* Can you please provide examples of generated molecules? The picture of molecular graphs and real drugs for given targets is fine.

* How did you make run docking process so fast? From my experience even GPU-accelerated programs for docking are quite slow. Is acceleration achieved because the docking is performed to the fixed target protein?

---

> ### Author Response · Authors · 2023-11-17
> **Response to AgdE (1  / 2)**
>
> Dear reviewer,
>
> We thank the reviewer for their feedback and summary of our work. We appreciate the reviewer’s assessment that the paper proposes a promising approach for molecular generation with extensive, relevant ablations. It seems like the reviewer’s main concerns are missing relevant work, presentation of the results, and examples of generated and hacked molecules. We have modified our paper to incorporate these concerns:
>
> 1) In Table 1 we add the relevant citations and a conceptual comparison with our work.
> 2) In Table 6,7,8,9 we add individual scores of the pretrained and RL trained agent on all PyTDC and docking tasks.
> 3) In Figure 14, we include molecules that are sampled as a result of reward hacking.
> 4) In Figure 18 - 40, we add the top 5 molecules corresponding to all 21 PyTDC tasks.
>
>  **Do these new updates and experiments, with our answers to specific questions, address all the reviewer's concerns?**
>
> > Some relevant works seem to be missing:
>
> We have added the first two papers (MolGPT, Taiga) in Table 1 of our updated paper, and can cite the third one as concurrent work after it is public.
>
> > The second work seems to be very similar in terms of approach
>
> Indeed, the second work is similar to ours in that both papers use a pre-trained transformer and train it using policy gradients, and we have added both papers in Table 1 of our updated paper. The difference is in the scope of our experiments and the large number of other design choices combined in a novel manner. For example,
> 1) Both the papers mentioned by you only show results for 2 or 3 different tasks, which include QED or SA scores. These tasks have been categorized as substantially easier to optimize in the literature compared to the tasks in our experiments [1], which span 25 + tasks including docking scores.
> 2) In previous papers, the size of the datasets used were around 1 million molecules. We conduct experiments with sizes up to 100 million molecules.
> 3) Moreover, we also study algorithmic components like regularization and the use of replay buffers which seem to have a huge impact on the results, but are not studied in the papers you mention.
> 4) Our experiments also include more architectural diversity, including fully-connected MLPs, RNNs and Transformer backbones.
>
> > I haven't noticed the comparison against any other methods.
>
> There are not many prior works that have been applied to text-based representations of molecules. Hence our goal was to extensively compare various design and algorithmic choices. However, many of our ablations can be recombined in different ways to recover prior algorithms. In Table 1, we show how many prior algorithms can be categorized by the design choices that we compare in our experiments.
>
> > What are "Div." and "Red." columns?
>
> These mean diversity and redundancy respectively. We have added the clarification in Section 5.1, evaluation metrics paragraph and also in the caption of Figure 2.
>
> > I couldn't find reported scores for different tasks (e.g., similarity, QED or SA).
>
> In Table 6,7,8,9, we have added the exact individual scores achieved by just the pretrained model and the benefits achieved from reinforcement learning on these pretrained models across all 26 docking and PyTDC tasks. These tables show that the RL algorithm improves the pretrained model by 69% and 35% on docking and PyTDC tasks respectively.
>
> > Reward hacking is mentioned by the authors but including resulting molecules is essential.
>
> In Figure 14 (Page 25) we have added examples of molecules sampled due to reward hacking. We can see that ChemRLformer agents are able to obtain unusually high docking scores by stacking together long chains and rings of sulfur, phosphorus or carbon atoms.
>
> Below, we will answer the specific reviewer questions and weaknesses. **Do these revisions and answers address all the reviewer's concerns?**
>
> >  What are the principal differences between your approach and Taiga?
>
> The largest difference is the scale of our experiments. Our experiments are much more extensive when compared to Taiga:
> 1) Num of tasks –  ours : 25, Taiga : 2.
> 2) Docking scores – ours : Yes, Taiga : No.
> 3) Largest datasets used – ours : 100 million, Taiga : 1 million.
> 4) Architectures compared – ours : FC, Transformer, RNN. Taiga : Transformers.
> 5) Important Algorithmic choices — ours : hill climb replay buffers, log p regularization. Taiga : None.
>
> We also have other contributions that differentiate us from prior work:
>
> 1) We show that current docking functions are not perfect, and can be easily hacked by ChemRLformer.
> 2) We show that when choosing a pre-training datasets, the alignment matters more than its size.
> 3) We show that transformers aren’t superior to RNNs for current string-based RL algorithms, at the scale of data currently available.

---

> > ### Author Response · Authors · 2023-11-17
> > **Response to AgdE (2 / 2)**
> >
> > > Can you compare against MolGPT, Taiga and REINVENT (or similar approach)?
> >
> > We have addressed the algorithmic differences between our approaches above. Overall, our work studies more ambitious settings and uncovers clear shortcomings in assessment methods, making detailed comparisons with additional methods challenging. We will add a empirical comparison with these methods in the camera-ready version.
> >
> > >  Can you please provide examples of generated molecules?
> >
> > In Figure 18 - 40, we add the top 5 molecules corresponding to all 21 PyTDC tasks.
> >
> > > How did you make run docking process so fast?
> >
> > We used quick vina two with multiprocessing to improve the speed of molecular docking. We also fix the protein target (for eg, Fa7, PARP1) at the start of the RL run. We have mentioned all the details as well as the exact speed of the docking simulation in Appendix A, paragraph “Docking tasks” and Table 3 in our paper. It took 130 seconds to dock 1000 molecules.
> >
> > [1] Ciepliński, Tobiasz, et al. "Generative Models Should at Least Be Able to Design Molecules That Dock Well: A New Benchmark." Journal of Chemical Information and Modeling (2023).

---

> > > ### Author Response · Authors · 2023-11-20
> > > **Rebuttal Follow-up**
> > >
> > > Dear Reviewer, We hope that you've had a chance to read our responses and clarification. As the end of the discussion period is approaching, we would greatly appreciate it if you could confirm that our updates have addressed your concerns.

---

> > > > ### Comment · Reviewer_AgdE · 2023-11-21
> > > >
> > > > I thank authors for their rebuttal work. My concerns are resolved. I've increased the score and confidence.

---

### Official Review · Reviewer_ANoo · 2023-10-30

**Soundness:** 3 good
**Presentation:** 3 good
**Contribution:** 3 good
**Rating:** 8
**Confidence:** 2

**Summary:**

The authors present a method for molecule design using language models and RL. First, a GPT-style model is pre-trained in an autoregressive fashion. The model is then finetuned using REINFORCE algorithm to generate high-value molecules. The authors discuss several design choices and perform ablation studies to confirm their hypotheses.

**Strengths:**

1. The problem is of high importance
1. The paper is well-written
1. The numerical results are well documented and contain ablation studies

**Weaknesses:**

1. Some of the design choices seem to be dated. For example,
    * it is not clear why REINFORCE is used as the policy learning algorithm, as opposed for instance to PPO.
    * there is no automatic entropy tuning for exploration.
    * Reinforcement Learning with Human Feedback can be also an interesting approach to try
1. The details for the backbone language model as scarce. It would be also interesting to discuss the design choices for language modeling, e.g., pretraining architecture (BERT vs GPT),  tokenizer etc
1. It would be good to see if increasing the transformer size would lead to performance improvement. If I am not mistaken increasing the RNN size could lead to issues while training and therefore this could potentially become the strength of the paper.

**Questions:**

1. A most popular approach nowadays to finetuning transformers is reinforcement learning with human feedback. In RLHF, the reward preference model is trained to choose between several samples of the transformer and then the fine-tuning is performed using PPO algorithm. Have the authors considered this approach?
1. It is not clear why the authors use Reinforce instead of more popular PPO for example. I appreciate that the authors did an ablation study on the KL constraint, but PPO has other benefits in comparison with Reinforce
1. Details of the backbone model are not given. Is it a standard GPT model? Was the tokenizer standard or also trained?
1. It is not clear why the authors chose GPT-style model as opposed to BERT style model, where the whole sequences can be predicted directly. This approach was used, e.g  by Cowen-Rivers et al albeit for a different problem and with a different RL algorithm. A discussion on the subject would be interesting.
1. Haarnoja et al 2018 proposed an automatic tuning procedure of entropy that can explicitly state the target level of entropy. It would be interesting to have this as ablation as well.
1. I am confused about referencing Mnih et al 2013 regarding the use of replay buffer for on-policy algorithms. Don’t Mnih et al use an off-policy algorithm?
1. Furthermore, I am not quite sure what a replay buffer for on-policy algorithm means. Does it mean that we use samples for several previous iterations not just the most recent one?
1. The results could be displayed better, with this scale the improvement of the model does not seem too large, but could be deceptive.
1. In Figure 2 CheMBL offers high diversity and high redundancy results, which seems confusing at first. Please comment in the figure caption

UPD:
References :
* Cowen-Rivers, Alexander I., et al. "Structured Q-learning For Antibody Design." arXiv preprint arXiv:2209.04698 (2022).
* Haarnoja, Tuomas, et al. "Soft actor-critic algorithms and applications." arXiv preprint arXiv:1812.05905 (2018).

---

> ### Author Response · Authors · 2023-11-17
> **Response to ANoo (1 / 2)**
>
> Dear reviewer,
>
> We thank the reviewer for their feedback. We appreciate the reviewer’s assessment that the paper is of high importance and well-written with extensive experiments and ablation studies. It appears that the reviewer's main concerns are the choice of using the REINFORCE algorithm, additional experiments, details of the model and presentation of results. We address the concerns using new experiments and updates to the paper:
>
> 1) Figure 13 compares REINFORCE with PPO across 21 tasks. This result supports our choice of using REINFORCE which achieves superior performance while being simpler.
>
> 2) We add Table 6,7,8,9 which contain individual scores of the pretrained and RL trained agent on all PyTDC and docking tasks.
>
> 3) We update appendix A to contain all details related to the experiments, including details of the size and hyperparameters of all pretraining models in appendix A.3, para “pre training models”.
>
> **Do these new experiments, with our answers to specific questions, address all the reviewer's concerns?**
>
> > It is not clear why the authors use Reinforce instead of more popular PPO for example
>
> Although PPO has been shown to generally be a better algorithm for control problems, the comparison is not clear in the case of molecular optimization ([1,2]) . We perform an experiment across 21 PyTDC objectives (see Figure 13, page 22) to compare REINFORCE with PPO. Our results indicate that vanilla policy gradient algorithms achieve higher performance on all metrics and are more stable when compared to actor critic algorithms like PPO. This experiment guided the choice of using REINFORCE over PPO.
>
> >  finetuning transformers is reinforcement learning with human feedback.
>
> While we agree that incorporating feedback from humans is a promising future direction, one reason why this will be more difficult in the space of molecular optimisation is that the humans that would be needed to give feedback are expert chemists. Given the scarcity of experts, the RLHF framework may not scale well for this use case. Indirectly, those experts are performing this already by creating simulations (e.g. Autodock) which we use in our paper.
>
> > The details for the backbone language model are scarce.
>
> In appendix A.3, in paragraph ‘pretraining models’ we have included the number of parameters and architecture specification of all the models used.
>
> > automatic tuning procedure of entropy that can explicitly state the target level of entropy
>
> The logp regularization we applied is similar to entropy regularization in that both regularize the log probability of the policy. Entropy regularization reduces log(prob), while logp regularisation increases 1 / log(prob). We add this clarification in the last paragraph of section 5.4
>
> > The details for the backbone language model
>
> In appendix A, we have included Table 4 and Table 5, which contains all the details on the pretraining models.
>
> > It would be good to see if increasing the transformer size helps
>
> We agree that this would be promising. But more importantly, our work uncovers an immediate problem of reward hacking of the docking functions that needs to be solved by domain experts. This is a problem that cannot be solved by increasing the transformer size alone. In addition, to make good use of a larger tranformer, we would also need larger, more diverse data, which is currently limited in the community.
>
> > In RLHF, the reward preference model
>
> Learning a reward preference model needs a large amount of expert human feedback. Rewards are limited in real-world chemistry, unlike general language chatbots. We need methods that use minimal rewards. Hence, we focus on more sample efficiency [1].
>
> > It is not clear why the authors use Reinforce instead of more popular PPO for example.
>
> We add a new Figure 13 (page 22) to compare REINFORCE with PPO across 21 molecular optimization tasks.  We found that for these tasks, more complicated approaches were more unstable than REINFORCE, which seems to perform better. These results resonate with other work in the molecular optimisation community [1,2]
>
> > I am confused about referencing Mnih et al 2013 regarding the use of replay buffer for on-policy algorithms
>
> You are correct in that Mnih et al 2013 use an off-policy algorithm, while we use REINFORCE which is on policy. We use replay buffers [3] to store high-scoring molecules and re-sample them to make REINFORCE updates, making the algorithm off-policy.  In appendix A.5, we add details in our paper about how this is done in practice.
>
> > Does it mean that we use samples for several previous iterations, not just the most recent one?
>
> Yes. We store a small number of previously seen high-scoring molecules and apply REINFORCE updates on them. Although this makes the algorithm slightly off policy, it vastly improves the performance (Figure 5). In appendix A.5, we add details in our paper about how this is done in practice.

---

> > ### Author Response · Authors · 2023-11-17
> > **Response to ANoo (2 / 2)**
> >
> > >  The results could be displayed better, with this scale the improvement of the model does not seem too large, but could be deceptive.
> >
> > In Table 6,7,8,9, we have added the exact individual scores achieved by just the pretrained model and the benefits achieved from reinforcement learning on these pretrained models across all 26 docking and PyTDC tasks. These tables show that the RL algorithm improves the pretrained model by 69% and 35% on docking and PyTDC tasks respectively.
> >
> > > In Figure 2 CheMBL offers high diversity and high redundancy results, which seems confusing at first. Please see the comment in the figure caption.
> >
> > We have linked the explanation in the caption of Figure 2 and added it in the main paper.  This is because diversity is measured using average pairwise Tanimoto similarity between Morgan fingerprints of the molecules, which can be high even if a small fraction of molecules are vastly different from the rest.
> >
> > [1] Ciepliński, Tobiasz, et al. "Generative Models Should at Least Be Able to Design Molecules That Dock Well: A New Benchmark." Journal of Chemical Information and Modeling (2023).
> >
> > [2] Gao, Wenhao, et al. "Sample efficiency matters: a benchmark for practical molecular optimization." Advances in Neural Information Processing Systems 35 (2022): 21342-21357.
> >
> > [3] Lin LJ. Self-improving reactive agents based on reinforcement learning, planning and teaching. Machine learning. 1992 May;8:293-321.

---

> > ### Comment · Reviewer_ANoo · 2023-11-18
> > **follow-up clarifications**
> >
> > Thank you for the detailed response and the new detailed experiments. You addressed many of my concerns, but before finalizing the assessment, I want to follow up with a few more clarifications:
> > 1. `Re: Reinforce vs PPO`. Could the authors provide the details for the Reinforce and PPO? Specifically, I am curious if the KL penalty, the replay buffer, and the entropy regularization were used in both algorithms. I am interested if these additions made a difference or if there's something inherent in this problem that makes Reinforce a better option.
> > 1. `Re: entropy tuning`. Perhaps there is a misunderstanding regarding automatic tuning. One could have a weight parameter $-\alpha \log(\pi)$ instead of unweighted entropy $-\log(\pi)$. The parameter $\alpha$ can be chosen through hyperparameter search or tuned automatically. If I understood correctly the current approach sets $\alpha = 1$, and the question is if you have considered automatic tuning or reweighting.
> > 2.  `Re: replay buffer`. I see your point, and I think my confusion came from this sentence:
> > `Although text-based  RL algorithms are trained on-policy, prior work has proposed using a replay buffer to improve performance [Mnih et al., 2013].` This seems to imply that Mnih et al proposed using a replay buffer for on-policy algorithms, which is not correct. I recommend rephrasing it.
> > 3. I recommend adding a quick discussion (perhaps in the appendix) regarding RLHF and why it's preferable to use other RL methods. Also, note that the reward model can potentially be trained using simulators, as well.
> >
> > PS. In the future, I recommend highlighting the changes in the PDF with a different color to make the evaluations a bit easier.

---

> > > ### Author Response · Authors · 2023-11-19
> > > **Response to ANoo**
> > >
> > > Dear Reviewer,
> > >
> > > Thank you for clarifying the remaining concerns, which we have addressed by adding a new experiment and revising the paper (details below). **Does this address the reviewer's remaining concerns?**
> > >
> > > > Could the authors provide the details for the Reinforce and PPO?
> > >
> > > To keep the evaluation fair, we compared both the algorithms without the replay buffer and without the KL penalty. Both algorithms used logp regularization. We made sure that both algorithms use the same pretrained policy. Before making the choice to continue with REINFORCE we also compared it with PPO extensively, and always found REINFORCE to perform better. We believe this occurs because PPO learns a value function which is a difficult task when the reward function is sparse. This is indeed the case for molecular optimization. The agent only gets non zero rewards at the last step of the episode, when the molecule is scored. This makes the value function learning highly biased. Additionally, transformers have also been found to be unstable for actor-critic algorithms [1]. We have added more details of the experiment and this explanation in the caption of Figure 13 (green text).
> > >
> > > > Re: entropy tuning.
> > >
> > > We apologize for the misunderstanding. Indeed we could regularize the entropy of the policy with a tuned coefficient. In Figure 41, we compared the entropy regularization with an automatically tuned coefficient with the Log P regulariser across all 21 PyTDC tasks.  We see that adding automatic tuning does not lead to much improvement. For the reasons explained in Section 5.4, from line 333, we believe that Log P regularization is significantly better for molecular optimization (It helps the agent get out of local optima.
> > >
> > > We note that Soft Actor Critic [Haarnoja et al 2018] proposed the automatic entropy tuning for continuous action spaces, while our tasks have discrete actions. Automatic entropy tuning needs to select a target entropy. We follow the Soft Actor Critc and use $-0.98 * \log(1 / |A|)$, where |A| is equal to the number of tokens in the vocabulary. We have also added this in the caption of Figure 41.
> > >
> > > > Re: replay buffer
> > >
> > > Thank you for pointing this out, we have removed the DQN citation and added the correct reference (see line 286 in green text).
> > >
> > > > I recommend adding a quick discussion (perhaps in the appendix) regarding RLHF
> > >
> > > In Appendix E (green color), we have added a new paragraph discussing the use of RLHF in chemistry. We have also added a line (see line 347 in green) in the future work section that mentions learning a reward model in a sample efficient way. Like we said in the initial response, learning a reward model requires a large number of labeled samples. And although unlimited samples could be obtained from the simulators, we know that these simulators aren’t perfect [3]. Our experiment showing reward hacking (Figure 4 for results and Figure 14  for images of hacked molecules) also supports this. Real chemists will only be able to give feedback on a limited number of molecules and hence we should focus on RL algorithms that are sample efficient.
> > >
> > > > In the future, I recommend highlighting the changes in the PDF with a different color to make the evaluations a bit easier.
> > >
> > > Thank you for your recommendation. The new changes that we have incorporated now are in green color to make evaluation easier.
> > >
> > > [1] Stabilizing Transformers for Reinforcement Learning
> > >
> > > [2] Soft Actor-Critic for Discrete Action Settings
> > >
> > > [3] Ciepliński, Tobiasz, et al. "Generative Models Should at Least Be Able to Design Molecules That Dock Well: A New Benchmark." Journal of Chemical Information and Modeling (2023).

---

> > > > ### Comment · Reviewer_ANoo · 2023-11-19
> > > > **Thank you for the clarifications!**
> > > >
> > > > Thank you for the quick response!
> > > >
> > > > A few final recommendations:
> > > >
> > > > 1. Please add the details for the PPO vs Reinforce comparison to the appendix (editing the figure caption would suffice). I think this discussion could be interesting.
> > > >
> > > > 2. Regarding the entropy for the discrete SAC this is the reference that introduces it:
> > > >
> > > > ```
> > > > @article{christodoulou2019soft,
> > > >   title={Soft actor-critic for discrete action settings},
> > > >   author={Christodoulou, Petros},
> > > >   journal={arXiv preprint arXiv:1910.07207},
> > > >   year={2019}
> > > > }
> > > > ```
> > > >
> > > > On a personal note, I never had any luck with the target entropy of $-0.98 \log(1/ |A|)$ in the discrete case, which seems to promote a purely random policy.

---

> > > > > ### Author Response · Authors · 2023-11-19
> > > > > **Response to ANoo**
> > > > >
> > > > > Thank you for your final recommendations. We have addressed by revising the paper (details below). **Does this address the reviewer's remaining concerns?**
> > > > >
> > > > > > Please add the details for the PPO vs Reinforce
> > > > >
> > > > > We have added the details in the caption of Figure 13 (green text).
> > > > >
> > > > > > Regarding the entropy for the discrete SAC
> > > > >
> > > > > We have also added the citation for discrete SAC in the caption of Figure 41 (green text).

---

> > > > > > ### Comment · Reviewer_ANoo · 2023-11-20
> > > > > > **Apologies for a late reply!**
> > > > > >
> > > > > > I lowered my confidence in the score but raised the overall score.
> > > > > >
> > > > > > I realized that I didn't add the references in my review. I updated the review to reflect these references. Feel free to have a look!

---

### Official Review · Reviewer_QKJh · 2023-11-07

**Soundness:** 2 fair
**Presentation:** 3 good
**Contribution:** 3 good
**Rating:** 6
**Confidence:** 2

**Summary:**

This paper explores the potential of reinforcement learning (RL) methods to discover new, high-value molecules and presents a new RL-based molecular design algorithm called ChemRLformer. The authors conduct extensive experiments and analysis to show that ChemRLformer achieves state-of-the-art performance while being more straightforward than prior work. The paper provides unique insights into the application of RL to molecular design and highlights the importance of careful search space structuring and algorithm design.

**Strengths:**

1. The paper addresses an important problem in the field of molecular design, which has significant implications for society. The potential of RL methods to discover new, high-value molecules could have a major impact on drug discovery, materials science, and other fields.

2. The paper presents a new RL-based molecular design algorithm called ChemRLformer, which achieves state-of-the-art performance while being more straightforward than prior work. The authors provide unique insights into the application of RL to molecular design and highlight the importance of careful search space structuring and algorithm design.

3. The paper is well-written and easy to understand, even for readers who may not be familiar with RL or molecular design. The authors provide clear explanations of their methods and results, and use visual aids to help illustrate their points.

4. The authors conduct extensive experiments and analysis using 25 molecule design tasks, including computationally complex protein docking simulations. They explore how different design choices for text grammar and algorithmic choices for training can affect an RL policy’s ability to generate molecules with desired properties.

**Weaknesses:**

1. It is confusing why the authors assume a discount rate of 1 in Section 4.1. Even with a finite trajectory, a discount rate smaller than 1 is not obligatory, it can also help. A discount rate of 1 can prevent the agent from learning and executing long-term tasks, as it won't appropriately discount long-term returns, leading it to prioritize immediate rewards and neglect long-term benefits. It is suggested to justify this setting.

2. REINFORCE is a very classic yet old algorithm. It is highly recommend to try more recent algorithms, e.g., TROP or PPO. I am not an exert in RL for molecular optimization, but I doubt whether it is the state-of-the-art RL algorithm in this field. For example, (i) due to its reliance on stochastic policies and simple optimization methods, REINFORCE can be more susceptible to getting stuck in local optima, making it less effective in complex and high-dimensional environments; (ii) due to its on-policy mechanism, REINFORCE requires a large number of samples to estimate gradients accurately. This can make it computationally expensive and slow for complex tasks and environments.

3. It would be more convincing if domain experts can help to judge the effectiveness of ChemRLformer from molecular's perspective.

**Questions:**

Please refer to Weaknesses.

---

> ### Author Response · Authors · 2023-11-17
> **Response to QKJh**
>
> Dear reviewer,
>
> We thank the reviewer for their feedback. We appreciate the reviewer’s assessment that the paper addresses an important problem with a new algorithm, is well-written and provides an extensive analysis. It appears that the reviewer's main concerns are regarding the discount factor, the choice of REINFORCE and a shortage of domain experts' perspective on the effectiveness of ChemRLformer. We have incorporated new updates Section 4 (footnote on page 5 explaining the use of discount factor 1) and experiments :
>
> 1) Figure 13 compares REINFORCE with PPO across 21 tasks PyTDC. This result supports our choice of using REINFORCE which achieves superior performance while being simpler.
>
> 2) In Appendix D, we added remarks by domain experts on ChemRLformer.
>
>  **Do these new experiments, with our answers to specific questions, address all the reviewer's concerns?** We address all of these concerns in detail below:
>
> > It is confusing why the authors assume a discount rate of 1 in Section 4.1
>
> The primary reason why we set the discount factor as 1 is related to the problem setting. During search, it is common to set a maximum length of the molecule (100 in our experiments) that the agent can sample. Hence, the MDP is a finite-length MDP. Rewards are only obtained once in an episode, when the agent samples the stop action or the length of the molecule hits the maximum length. Having a discount factor of 1 is an intentional choice because discount factors less than one bias the agent towards molecules with shorter lengths. This bias is undesirable in practice.
>
> > REINFORCE is a very classic yet old algorithm.
>
> REINFORCE is a very classic yet old algorithm. It is highly recommended to try more recent algorithms, e.g., TROP or PPO. Although PPO has generally been shown to be a better algorithm for control problems, the comparison is not clear in the case of molecular optimization ([1,2]) . We perform an experiment across 21 PyTDC objectives (see Figure 13) to compare REINFORCE with PPO. Our results indicate that vanilla policy gradient algorithms achieve higher performance on all metrics and are more stable when compared to actor-critic algorithms like PPO. This experiment guided the choice of using REINFORCE over PPO.
>
> > It would be more convincing if domain experts can help to judge the effectiveness of ChemRLformer from molecular's perspective.
>
> We include comments of a Domain expert who has read the paper: They find the versatility of ChemRLformer in solving multiple tasks, especially expensive tasks like docking, useful for early-stage drug discovery. The reward-hacking insights are particularly useful as they show the shortcomings of docking score evaluations and the need for better evaluation, both in simulation and real-world experiments.
>
> We will add a deeper discussion on domain expert and application perspectives in the camera-ready version.
>
>
> [1] Ciepliński, Tobiasz, et al. "Generative Models Should at Least Be Able to Design Molecules That Dock Well: A New Benchmark." Journal of Chemical Information and Modeling (2023).
>
> [2] Gao, Wenhao, et al. "Sample efficiency matters: a benchmark for practical molecular optimization." Advances in Neural Information Processing Systems 35 (2022): 21342-21357.

---

> > ### Author Response · Authors · 2023-11-20
> > **Rebuttal Follow-up**
> >
> > Dear Reviewer,
> > We hope that you've had a chance to read our responses and clarification. As the end of the discussion period is approaching, we would greatly appreciate it if you could confirm that our updates have addressed your concerns.

---

### Meta-Review · Area_Chair_rt7b · 2023-12-07

**Metareview:**

The paper introduces ChemRLformer, a molecular optimization technique that uses RL with a GPT-like backbone model. After the author reviewer discussion phase, all the reviewers agreed that the paper provided unique insights (e.g. surprisingly REINFORCE with off-policy updates outperforms PPO across a variety of molecular design tasks), novel combinations of approaches (e.g. regularization and the use of replay buffers) and can inform the problem framing for future works on molecule design tasks.

**Justification For Why Not Higher Score:**

The primary weakness of the paper is the related work: reviewers pointed out other off-policy and on-policy RL methods using transformers for drug discovery and related tasks. The authors clarified that one of the related papers is concurrent work, but can do better to differentiate and position ChemRLformer's contributions vs. the other related works (including benchmarking them as baselines in their comprehensive evaluation).

**Justification For Why Not Lower Score:**

Several of the reviewers' concerns about the use of REINFORCE vs. state of the art RL techniques, experiment details, and ablations were adequately addressed by the authors during the feedback phase.

---

### Decision · Program_Chairs · 2024-01-16

Accept (poster)